# Pericentrin-mediated SAS-6 recruitment promotes centriole assembly

Daisuke Ito[1]*, Sihem Zitouni[1†], Swadhin Chandra Jana[1†], Paulo Duarte[1], Jaroslaw Surkont[1], Zita Carvalho-Santos[1‡], José B Pereira-Leal[1,2], Miguel Godinho Ferreira[1,3], Mónica Bettencourt-Dias[1]*

[1]Instituto Gulbenkian de Ciência, Oeiras, Portugal; [2]Ophiomics, Precision Medicine, Lisboa, Portugal; [3]Institute for Research on Cancer and Aging of Nice (IRCAN), INSERM U1081 UMR7284 CNRS, Nice, France

**Abstract** The centrosome is composed of two centrioles surrounded by a microtubule-nucleating pericentriolar material (PCM). Although centrioles are known to regulate PCM assembly, it is less known whether and how the PCM contributes to centriole assembly. Here we investigate the interaction between centriole components and the PCM by taking advantage of fission yeast, which has a centriole-free, PCM-containing centrosome, the SPB. Surprisingly, we observed that several ectopically-expressed animal centriole components such as SAS-6 are recruited to the SPB. We revealed that a conserved PCM component, Pcp1/pericentrin, interacts with and recruits SAS-6. This interaction is conserved and important for centriole assembly, particularly its elongation. We further explored how yeasts kept this interaction even after centriole loss and showed that the conserved calmodulin-binding region of Pcp1/pericentrin is critical for SAS-6 interaction. Our work suggests that the PCM not only recruits and concentrates microtubule-nucleators, but also the centriole assembly machinery, promoting biogenesis close by.

DOI: https://doi.org/10.7554/eLife.41418.001

*For correspondence:
itodaisuke1982@gmail.com (DI);
mdias@igc.gulbenkian.pt (MóB-D)

†These authors contributed equally to this work

Present address:
‡Champalimaud Centre for the Unknown, Lisbon, Portugal

Competing interests: The authors declare that no competing interests exist.

## Introduction

The centrosome is the major microtubule organizing center found in animals, being composed of two microtubule-based cylinders, the centrioles, which are surrounded by an electron-dense protein-aceous pericentriolar material (PCM) that nucleates microtubules. The centriole can also function as a basal body, nucleating cilia formation. Centrioles are normally formed in coordination with the cell cycle, one new centriole forming close to each existing one, within a PCM cloud (*Bettencourt-Dias and Glover, 2007*; *Fu et al., 2015*).

The first structure of the centriole to be assembled is the cartwheel, a ninefold symmetrical structure, composed of SAS-6, CEP135/Bld10, STIL/Ana2/SAS-5, amongst others (*Kitagawa et al., 2011*; *Lin et al., 2013*; *Nakazawa et al., 2007*; *Tang et al., 2011*; *van Breugel et al., 2011*). This is followed by centriole elongation through the deposition of centriolar microtubules which is dependent on components such as CPAP/SAS-4 (*Kohlmaier et al., 2009*; *Schmidt et al., 2009*; *Tang et al., 2009*). Remarkably, SAS-6 and CEP135/Bld10 can self-assemble in vitro to generate a ninefold symmetrical cartwheel (*Guichard et al., 2017*). In vivo, active PLK4 is known to recruit and phosphorylate STIL, which then recruits SAS-6 (*Arquint et al., 2012*; *Arquint et al., 2015*; *Moyer et al., 2015*; *Ohta et al., 2014*). In a process called 'centriole-to-centrosome conversion', the centriole recruits centriole-PCM linkers, such as CEP192/SPD2 and CEP152/asterless, which contribute to both centriole biogenesis and PCM recruitment (*Cizmecioglu et al., 2010*; *Dzhindzhev et al., 2010*; *Hatch et al., 2010*; *Sonnen et al., 2013*). Finally, CDK5RAP2/CNN and pericentrin are recruited, which then recruit and activate γ-tubulin and create an environment favorable for concentrating

tubulin, leading to the formation of a proficient matrix for microtubule nucleation and organization (*Delaval and Doxsey, 2010*; *Feng et al., 2017*; *Megraw et al., 2011*; *Woodruff et al., 2017*).

The cascade of phosphorylation and interaction events between centriole components leading to centriole biogenesis is an intricate succession of positive feedback interactions. That circuit leads to amplification of an original signal present at the centriole, such as the presence of active PLK4, or its substrate STIL, hence perpetuating centriole biogenesis there (*Arquint and Nigg, 2016*; *Lopes et al., 2015*; *Moyer et al., 2015*). Additionally, the already existing 'older' centriole is surrounded by PCM, which could help to localize and concentrate centriole components during centriole assembly, reinforcing centriole formation close by. In support of this idea, when centrioles are eliminated in human cells by laser ablation, they form de novo within a PCM cloud (*Khodjakov et al., 2002*). Moreover, exaggeration of the PCM cloud by overexpressing a PCM component, pericentrin, in S-phase-arrested CHO cells, induces the formation of numerous daughter centrioles (*Loncarek et al., 2008*). Although pericentrin is not shown to be implicated in centriole biogenesis per se to date, dissociation of another pericentrin-related protein, AKAP450, from the centrosome, interferes with centriole duplication in human cells (*Keryer et al., 2003*). Moreover, PCM components, γ-tubulin and Spd-5, a functional analogue of *Drosophila* CNN and Hs Cdk5rap2, contribute to centriole assembly in *C. elegans* (*Dammermann et al., 2004*). Downregulation of the PCM in *Drosophila,* has been shown to lead to: i) fragmented and short centrioles, in the case of pericentrin (PLP) (*Martinez-Campos et al., 2004*; *Roque et al., 2018*), ii) disengaged centrioles, in the case of CNN (*Lucas and Raff, 2007*; *Megraw et al., 2001*), and iii) centriole disassembly, in the case of the removal of all PCM components (*Pimenta-Marques et al., 2016*). It is thus likely that the PCM plays critical roles in assembling and/or maintaining centriole structures. Despite the possible importance of the PCM, it is difficult to study its roles on centrioles due to confounding effects of the many centriole-centriole component interactions that exist in centriole-containing animal cells.

To define the contribution of PCM in regulating centriole components, we explored a system that has no endogenous centriole components, taking advantage of the diverged centrosomes observed in nature. While centrioles are ancestral in eukaryotes, they were lost in several branches of the eukaryotic tree, concomitantly with the loss of the flagellar structure (*Carvalho-Santos et al., 2010*). Instead of a canonical centrosome, yeasts have a spindle pole body (SPB), a layered structure composed of a centriole-less scaffold that similarly recruits γ-tubulin and other PCM components and nucleates microtubules (*Cavanaugh and Jaspersen, 2017*). The timing and regulation of SPB biogenesis are similar to the one observed in animal centrosomes (*Lim et al., 2009*; *Kilmartin, 2014*; *Rüthnick and Schiebel, 2016*). It is likely that the animal centrosome and yeast SPB evolved from a common ancestral structure that had centrioles (*Figure 1A*), as early-diverged basal fungi such as chytrids (e.g. *Rhizophydium spherotheca*) have a centriole-containing centrosome (*Powell, 1980*). By expressing animal centriole components in fission yeast, we observed that the SPB recruits them. We further demonstrate that the SPB conserved PCM component Pcp1/pericentrin recruits the centriole component SAS-6. We further validated this interaction in animals and show it is important for centriole elongation. Our work reveals an important role for pericentrin in recruiting centriole components and regulating centriole structure in animals.

## Results

We first investigated the conservation of centrosome components, searching for orthologues of the known animal proteins comprising: components of centrioles that are required for centriole biogenesis (SAS-6, STIL/Ana2/SAS-5, CPAP/SAS-4, CEP135/Bld10 and CEP295/Ana1), linkers of the centriole to the PCM, which are bound to the centriole and are required for PCM recruitment (CEP152 and CEP192), and the PCM itself, which is involved in γ-tubulin recruitment and anchoring (pericentrin, Cdk5rap2 and γ-tubulin itself). In addition, to better understand when the SPB originated, we also searched for orthologs of the fission yeast SPB components: the core scaffold proteins (Ppc89, Sid4, Cdc11, Cut12 and Cam1; *Bestul et al., 2017*; *Chang and Gould, 2000*; *Krapp et al., 2001*; *Moser et al., 1997*; *Rosenberg et al., 2006*) and the half-bridge proteins (Sfi1 and Cdc31; *Kilmartin, 2003*; *Paoletti et al., 2003*), which are required for SPB duplication.

Consistent with previous studies (*Carvalho-Santos et al., 2010*; *Hodges et al., 2010*), the proteins required for centriole biogenesis in animals were not identified in the fungal genomes, with exception of chytrids, which have centrioles (*Figure 1B*). Centriole-PCM connectors were not found

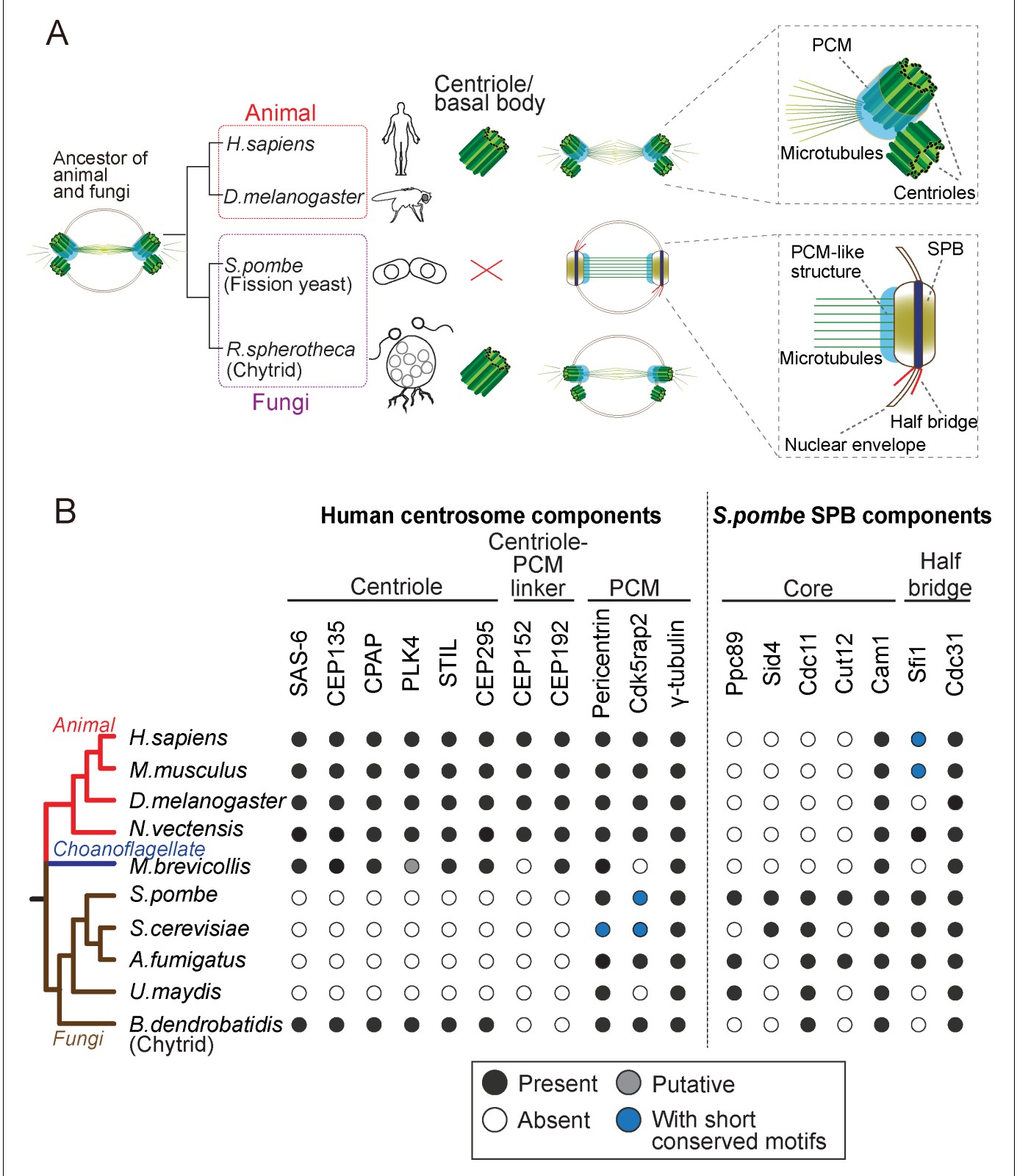

**Figure 1.** Evolution of the morphology and protein content of animal and fungi centrosomes. (**A**) The structure of the centrosome in mitosis in most animals, chytrids (flagellated fungi) and fission yeast. Animals and chytrids have a centriole/basal body and a canonical centrosome composed of a pair of centrioles surrounded by PCM, which anchors and nucleates microtubules. Fission yeast lacks a centriole but has a spindle pole body (SPB) inserted in the nuclear envelope. The SPB nucleates microtubules from the PCM-like structure, inside the nucleus. Parsimoniously, it is likely that the common

*Figure 1 continued on next page*

*Figure 1 continued*

ancestor of animals and fungi had a centriole-containing centrosome with a PCM structure (model shown). (**B**) Phylogenetic distribution of centrosome components in opisthokonts (animals, fungi and choanoflagellates). We searched for orthologues of components of the human centrosome localizing to centrioles, centriole-PCM linkers and PCM, and the fission yeast SPB components. Black circles represent the presence of orthologues that were identified by the bidirectional best hit approach to the human or fission yeast proteins, respectively; gray circle represents the presence of a putative orthologue identified by constructing phylogenetic trees; blue circles indicate that previous studies showed the presence of a protein with short conserved motifs (*Kilmartin, 2003*; *Samejima et al., 2010*; *Lin et al., 2014*) although we failed to identify it by the computational methods highlighted above; white circles indicate no detectable orthologue.

DOI: https://doi.org/10.7554/eLife.41418.002

The following source data is available for figure 1:

**Source data 1.** List of the predicted orthologues of the human centrosome and *S. pombe* SPB components in animal and fungi species.
DOI: https://doi.org/10.7554/eLife.41418.003

in both chytrids and yeasts. In contrast, when investigating the PCM composition, pericentrin and Cdk5rap2 were found in all fungal species, although the budding yeast Spc110 and Spc72 only share short conserved motifs with pericentrin and Cdk5rap2, respectively (*Lin et al., 2014*).

Regarding the SPB components, we found that Cam1 and Cdc31 are both highly spread across opisthokonts, which may reflect a conserved module or, alternatively an MTOC-independent conserved function of these proteins. On the other hand, we could only find proteins such as Ppc89 and Sid4 in yeasts, but not in chytrids, suggesting that some of the yeast-specific structural building blocks of the SPB appeared after branching into yeasts (*Figure 1B*).

Altogether, these results suggest the yeast centrosome is very different from the animal canonical centrosome, not having centriole and centriole-PCM adaptors, and being composed of several yeast-specific SPB components. Our results suggest that different proteins are involved in assembling different modules of the centrosome and that they can be lost when that module is lost, leading to a divergence of the remaining structures. Importantly, while centriole components were lost, the PCM module is conserved in animals and fungi in terms of composition and function, establishing fission yeast as an interesting system to study the interaction between components of the centriole and the PCM. We thus expressed key animal centriole components in fission yeast and asked whether they would interact with the PCM at the SPB or other yeast microtubule organizing centers.

## Animal centriole components localize to the fission yeast SPB

We used individual *Drosophila* centriole components as they are well-characterized and tested their localization when expressed in fission yeast. We chose five critical components, SAS-6, CEP135/Bld10, CPAP/SAS-4, STIL/Ana2/SAS-5 and PLK4 for the test. All genes coding for these proteins are absent from the yeast genomes (*Figure 1B*). Fission yeast SPBs are easily recognizable under light microscopy with fluorescent protein-tagged SPB marker proteins, such as Sid4 and Sfi1 (*Chang and Gould, 2000*; *Kilmartin, 2003*), which show distinct localization as a clear focus. Therefore, we examined if the animal centriole proteins could recognize and thus localize to the SPB.

GFP or YFP-tagged *Drosophila* centriole proteins were heterologously expressed under control of the constitutive *atb2* promoter (*Matsuyama et al., 2008*) or the inducible *nmt1* promoter (*Maundrell, 1990*) in fission yeast. Despite the one billion years separating yeasts from animals (*Douzery et al., 2004*; *Parfrey et al., 2011*), SAS-6-GFP, Bld10-GFP and YFP-SAS-4 co-localized with fission yeast Sid4 to the SPB (*Figure 2A and B*). In addition to the localization on the SPB, YFP-SAS-4 signal was also weakly observed along interphase cytoplasmic microtubules, likely reflecting its microtubule-binding capacity (*Gopalakrishnan et al., 2011*). In contrast, GFP-Plk4 and YFP-Ana2 did not localize to the SPB and existed as foci in the cytoplasm (*Figure 2A and B*). We confirmed the expression of the fusion proteins with the expected sizes (*Figure 2—figure supplement 1A*). Cells expressing centriole proteins which localize to the SPB grew as well as control cells, which have no centriole protein (*Figure 2—figure supplement 1B*), suggesting their expression does not impair yeast growth.

The surprising result that SAS-6, Bld10 and SAS-4 independently localize to the SPB, suggests that one or more fission yeast SPB component can recruit them. These results indicate that SPBs and canonical centrosomes have diverged less than what would be expected from their diverse morphology and divergent protein composition.

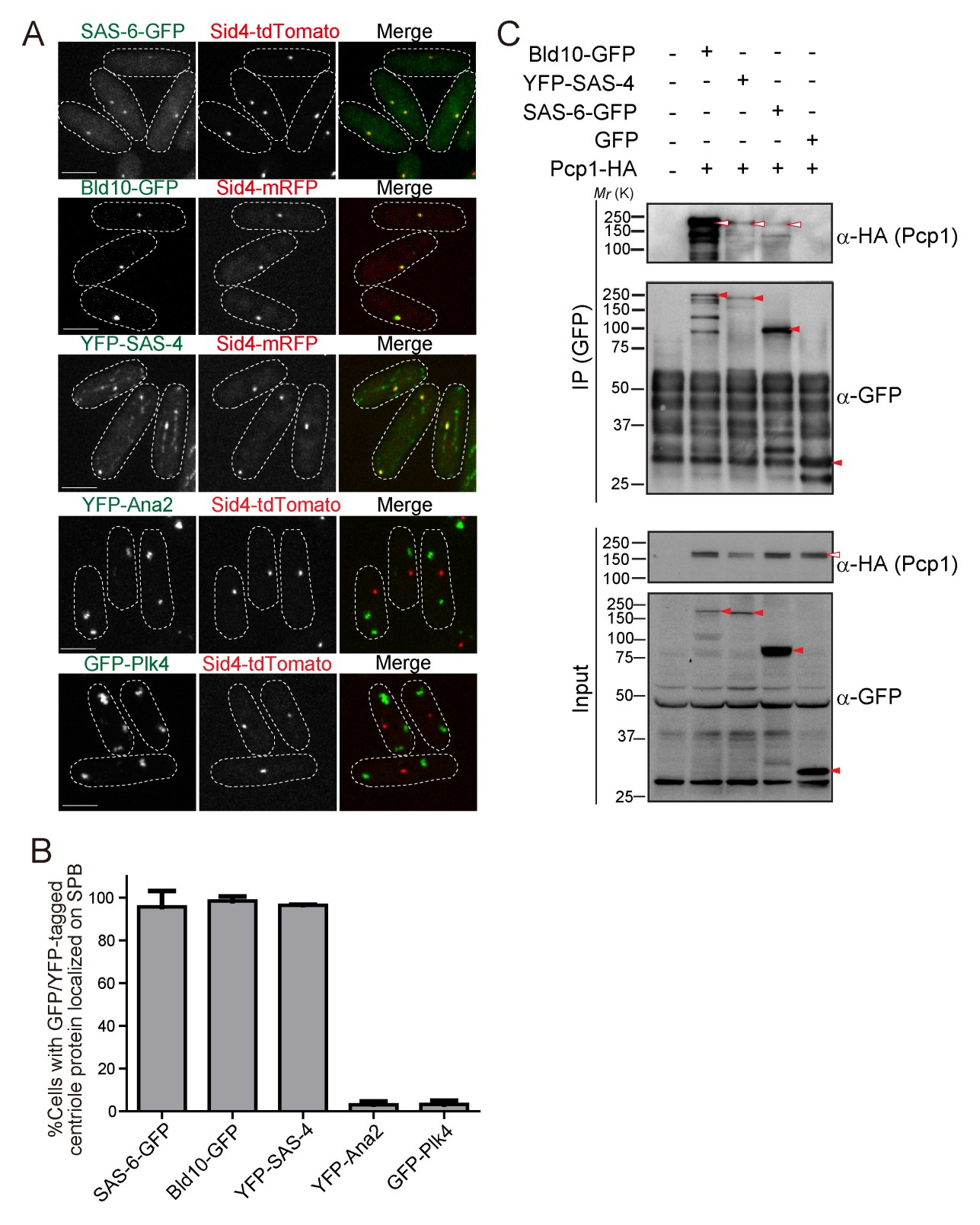

**Figure 2.** *Drosophila* centriole proteins localize to the centrosome of fission yeast. (**A**) SAS-6, Bld10 and SAS-4 localize to the SPBs, while Ana2 and Plk4 do not (see Materials and methods for details on expression constructs). Scale bar, 5 µm. (**B**) Quantification of cells with GFP or YFP-tagged centriole proteins localized on the SPB. Data are the average of three experiments ± s.d. (N > 50, GFP/YFP-positive cells). (**C**) Physical interaction between the centriole proteins and fission yeast Pcp1. Protein extract was prepared from fission yeast cells expressing HA-tagged Pcp1 and either SAS-6-GFP, or

*Figure 2 continued on next page*

*Figure 2 continued*

Bld10-GFP, or YFP-SAS-4 or GFP. The GFP-tagged proteins were immunoprecipitated with anti-GFP antibody. Immunoprecipitates and inputs (4%) were analyzed by western blotting using the indicated antibodies. Red open and filled arrowheads indicate the bands of Pcp1-HA and GFP/YFP fusion proteins, respectively.

DOI: https://doi.org/10.7554/eLife.41418.004

The following source data and figure supplement are available for figure 2:

**Source data 1.** The source data to plot the graph in *Figure 2B*.

DOI: https://doi.org/10.7554/eLife.41418.006

**Figure supplement 1.** Expression of *Drosophila* centriole proteins in fission yeast and the growth of fission yeast strains.

DOI: https://doi.org/10.7554/eLife.41418.005

## Fission yeast pericentrin, Pcp1, interacts with centriole proteins

Given the conservation of PCM components (*Figure 1B*), we wondered whether fission yeast PCM components localized on the SPB could be recruiting the *Drosophila* centriole components. We first examined the interaction between the centriole proteins (SAS-6, Bld10 and SAS-4) and Pcp1, the fission yeast pericentrin ortholog, which recruits the γ-tubulin ring complex (γ-TuRC) to regulate mitotic spindle formation (*Fong et al., 2010*). In animals, pericentrin is a key component of the PCM, extending with its C-terminus at the centriole wall into the PCM (*Lawo et al., 2012*; *Mennella et al., 2012*). Surprisingly, we found that SAS-6, Bld10 and SAS-4, all interacted with Pcp1 as revealed by co-immunoprecipitation (*Figure 2C*).

## Fission yeast Pcp1 is required for SAS-6 recruitment

Next, we examined if Pcp1 is required for localization of SAS-6, Bld10 and SAS-4 on the SPBs using a temperature-sensitive mutant of Pcp1 (*pcp1-14*), in which the amount of Pcp1 protein is already reduced, and further reduced when grown at the restrictive temperature (*Tang et al., 2014*). These cells also arrest in mitosis when at the restrictive temperature. To compare the signal intensity in cells at the same cell cycle stage, mitosis, we introduced the *cut7-446 allele* both in wild-type and *pcp1-14* background. Cut7 is a mitotic kinesin, its mutation fails in interdigitating the mitotic spindle and causes the cells to arrest in early mitosis (*Hagan and Yanagida, 1990*). Hereafter, we refer to the strains with *cut7-446* and *pcp1-14 cut7-446* alleles, as the control and *pcp1* mutant, respectively. The intensity of SAS-6-GFP per SPB was significantly increased in control cells when arrested in prometaphase (*Figure 3A and B*). We observed reduced SAS-6-GFP intensity in the *pcp1* mutant both at the permissive and restrictive temperatures (*Figure 3A and B*). This indicates that Pcp1 is required for SAS-6 recruitment to the SPB. Unlike SAS-6, Bld10-GFP intensity was not reduced, but slightly increased in the *pcp1* mutant (*Figure 3C and D*). It is possible that Bld10 is recruited to the SPB by another SPB component(s) which is upregulated in the *pcp1* mutant. Although the intensity of SAS-4 was reduced in the *pcp1* mutant (*Figure 3E and F*), we found that the total protein of YFP-SAS-4 was also lower in the *pcp1* mutant while that of SAS-6-GFP and Bld10-GFP was comparable in control and *pcp1* mutant lysates (*Figure 3G*). We think that YFP-SAS-4 might be stabilised by Pcp1 in fission yeast cells, and therefore we cannot conclude whether Pcp1 is required for YFP-SAS-4 localization on the SPB. Since SAS-6 is such a critical component in centriole assembly, and its localization is determined by Pcp1, we decided to explore further how SAS-6 is recruited to the SPB by Pcp1.

It is known in *Drosophila* and human cells that phosphorylation and interaction of PLK4 with STIL/Ana2, facilitates STIL recruitment to the centriole and its interaction and recruitment of SAS-6 (*Arquint et al., 2012*; *Moyer et al., 2015*; *Ohta et al., 2014*; *Vulprecht et al., 2012*). However, neither Plk4 nor STIL are present in the fission yeast genome (*Figure 1B*), suggesting that Pcp1/pericentrin is part of an additional and ancestral molecular pathway for SAS-6 recruitment.

## SAS-6 interacts with the conserved region of Pcp1, and Pcp1 is sufficient to recruit SAS-6

We reasoned that if the interaction between SAS-6 and Pcp1/pericentrin is an ancient and conserved connection, they should interact through an evolutionarily conserved domain in Pcp1. Subsequently, we determined which part of Pcp1 is required for its interaction with SAS-6. Full-length and

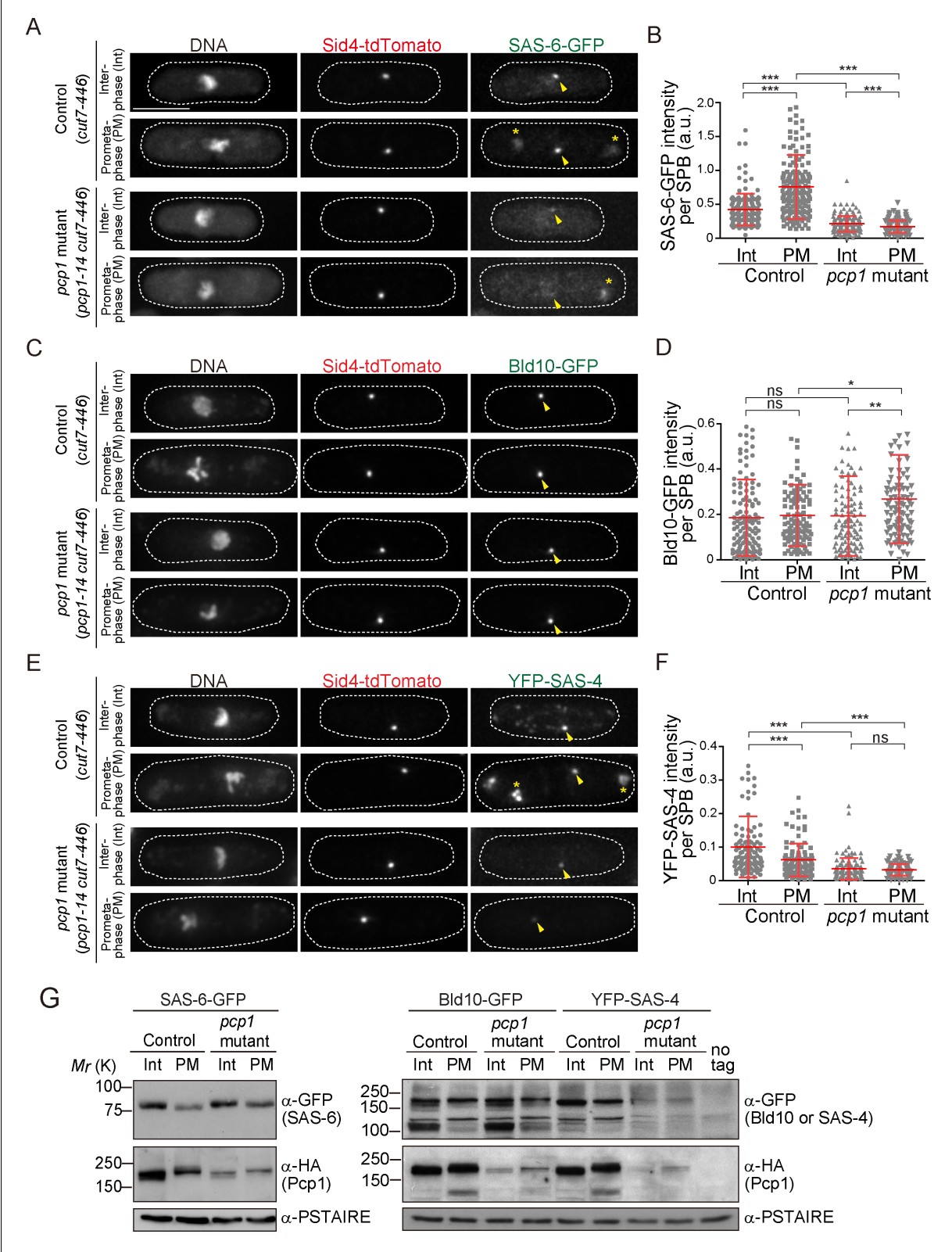

**Figure 3.** Fission yeast pericentrin-like protein Pcp1 is required to recruit SAS-6 to the SPB. (**A, C, E**) SAS-6-GFP, Bld10-GFP and YFP-SAS-4 intensities on the SPB in the *pcp1* mutant in asynchronous and prometaphase-arrested cells. The *cut7-446* (labeled 'control') and *cut7-446 pcp1-14* (labeled '*pcp1* mutant') strains expressing SAS-6-GFP and Sid4-tdTomato were incubated at the restrictive temperature (36°C) for three hours to block cells in prometaphase due to the *cut7-446* mutation (temperature-sensitive allele of a mitotic kinesin, which causes failure in mitotic spindle formation). *Figure 3 continued on next page*

*Figure 3 continued*

Representative images of the cells collected before shifting the temperature (interphase, Int) and three hours after the shift to 36˚C (prometaphase, PM, restrictive temperature) in control and *pcp1* mutant are shown. DNA was stained with DAPI. Arrowheads indicate the signal on the SPB. Note that we also observed aggregation of SAS-6-GFP and YFP-SAS-4 in the cytoplasm both in the control and *pcp1* mutant in all the cells exposed to the restrictive temperature (indicated with an asterisk), and never at the normal and permissive culture condition. We think that aggregate formation might stem from the *cut7-446* genetic background and/or the heat stress. Scale bar, 5 µm. (**B, D, F**) Quantification of the intensity of the centriole proteins per SPB in the indicated conditions. Means ± s.d. are shown in red (N > 100 SPBs, ns-not significant, *p<0.05, **p<0.001, ***p<0.0001, Mann-Whitney U test). (**G**) Western blotting analysis of protein extracts prepared from the indicated conditions.
DOI: https://doi.org/10.7554/eLife.41418.007
The following source data is available for figure 3:

**Source data 1.** The source data to plot the graphs in *Figure 3B,D,F*.
DOI: https://doi.org/10.7554/eLife.41418.008

truncation mutants of Pcp1 (N, M and C) were co-expressed with SAS-6-GFP (*Figure 4A*). Only full-length Pcp1 and its C-terminal region containing the conserved PACT domain interacted strongly with SAS-6 (*Figure 4B*). Given that the Pcp1-M fragment was weakly expressed, we cannot exclude completely the possibility that it also interacts with SAS-6-GFP (*Figure 4B*). The PACT domain localizes to the centriole wall in animals and is required for MTOC targeting both in animal and fungi (*Gillingham and Munro, 2000*).

We next asked whether Pcp1 could recruit SAS-6 ectopically. It is known that Pcp1 overexpression forms multiple Pcp1-containing foci in the cytoplasm (*Flory et al., 2002*). We thus examined if the cytoplasmic Pcp1 foci recruit SAS-6 (illustrated in *Figure 4C*). Overexpressed mCherry-tagged Pcp1 recruited SAS-6-GFP, but not GFP (control), to such foci (*Figure 4D and E*), indicating that Pcp1 can ectopically recruit SAS-6.

Though epifluorescence micrographs suggest SAS-6 co-localizes with Sid4 and Pcp1 (*Figures 2* and *4*), due to the resolution limit of a conventional optical microscope (~200 nm), we failed to conclude precisely where SAS-6 localizes to. To further determine the precise localization of SAS-6, we analyzed the relative position of SAS-6-GFP with respect to core SPB components Pcp1-tdTomato or Sid4-tdTomato (*Figure 4—figure supplement 1A*) by structured illumination microscopy (SIM). The foci of Pcp1-tdTomato and Sid4-tdTomato within the duplicated SPBs were distinguishably separated (60 ± 10 nm) (*Figure 4—figure supplement 1B and C*). Importantly, SAS-6-GFP center of mass localizes at ~50 nm distance from both Pcp1-tdTomato and Sid4-tdTomato center of mass. Given that the diameter and height of SPB is 180 nm and 90 nm respectively (*Ding et al., 1997*), our data suggest that ectopically expressed SAS-6-GFP localizes to the core of the SPB.

## Regulation of SAS-6 localization to the SPB

To further understand how SAS-6 is recruited to the SPB, we asked which of its domains are required for that localization. It has been reported that the coiled-coil region of human SAS-6 is sufficient for its localization to the animal centriole. This process is mediated by its interaction with other centriole components, such as STIL and CEP135 (*Keller et al., 2014*; *Moyer et al., 2015*). We expressed full-length SAS-6 and truncation mutants N-terminal (aa 1–176) and C-terminal (aa 177–472) and asked which co-localized with the SPB marker Sid4 (*Figure 4—figure supplement 2A*). Similar to what happens in animals, the C-terminal coiled-coil region of SAS-6 is required for its localization at the SPB (*Figure 4—figure supplement 2B,C and D*).

To approach the timing of SAS-6 targeting to the SPB, we asked when it would be targeted during the cell cycle. Firstly, we confirmed that SAS-6 expression under atb2 promoter was constant throughout the cell cycle in synchronized cells (*Figure 4—figure supplement 3A and B*). We observed that SAS-6 accumulated at the SPB before entering mitosis, similar to its described regulation in animals, and also similar to the reported recruitment of Pcp1 to the SPB at that cell cycle stage in fission yeast (*Keller et al., 2014*; *Strnad et al., 2007*; *Wälde and King, 2014*).

## Conservation of the Pcp1/pericentrin - SAS-6 interaction

Our experiments suggest that there are conserved interactions between centrioles and pericentrin, whose evolution is constrained. They further suggest that pericentrin has an important role in

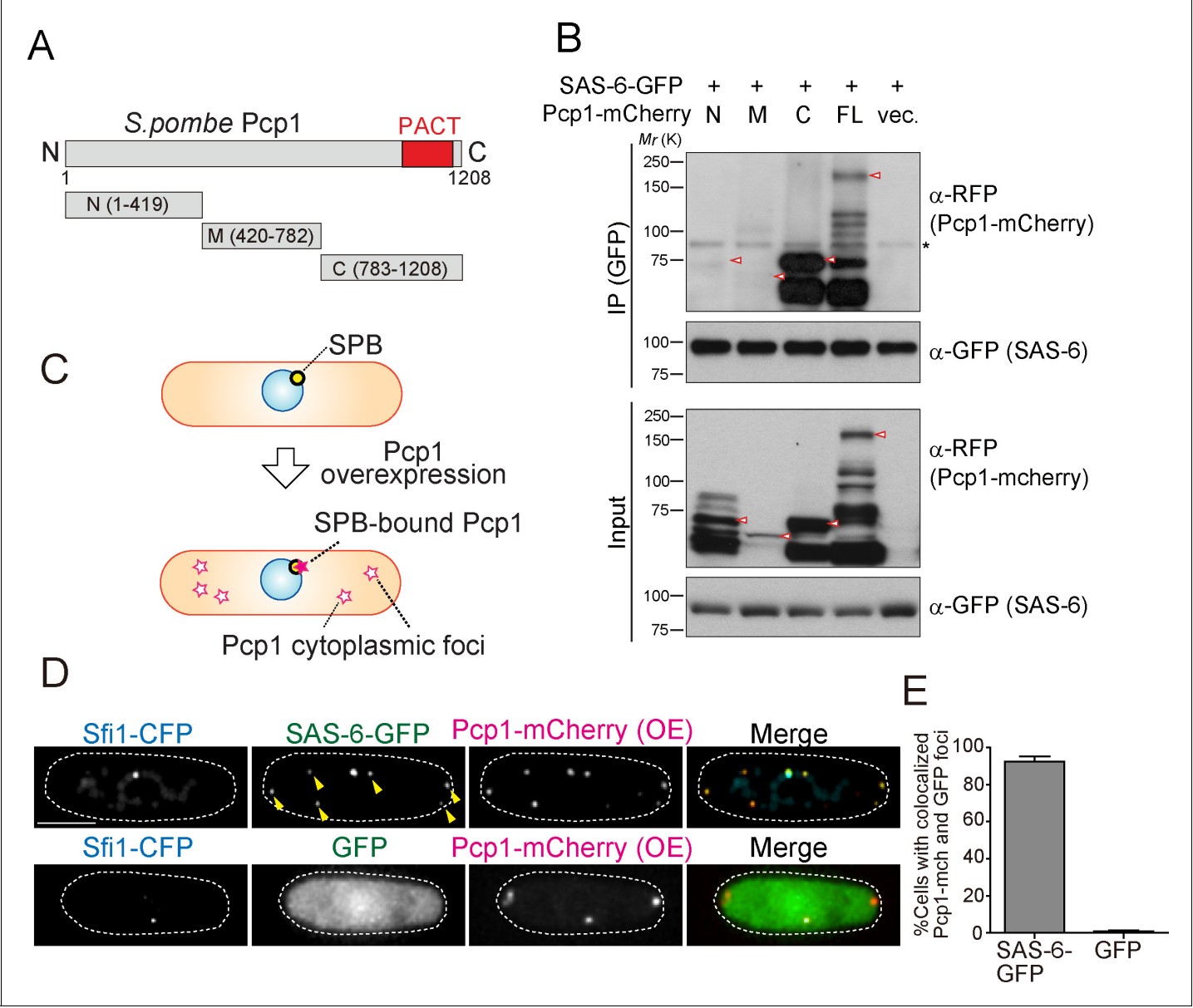

**Figure 4.** SAS-6 interacts with Pcp1 through the conserved carboxyl-terminal region, and Pcp1 is sufficient for SAS-6 localization. (A) Schematic illustration of the truncation constructs of Pcp1. (B) Pcp1 interacts with SAS-6 through the conserved carboxyl-terminal region. mCherry-tagged full-length or truncation mutants of Pcp1 were expressed in cells constitutively expressing SAS-6-GFP. Immunoprecipitation was performed and analyzed similarly as in *Figure 2B*. Red arrowheads indicate bands with the expected size of each fusion protein. The asterisk indicates non-specific bands. (C, D) Pcp1 is sufficient to recruit SAS-6. Overexpression of Pcp1 leads to the formation of Pcp1 containing cytoplasmic foci (schematic illustration, (C). Pcp1-mCherry was overexpressed under control of *nmt41* promoter in the strain expressing SAS-6-GFP and GFP alone. Sfi1-CFP is shown (SPB marker). Arrowheads indicate the SAS-6-GFP signal on the Pcp1-cytoplasmic foci. Scale bar, 5 μm. (E) Quantification of the cells with colocalized Pcp1-mCherry and GFP foci. Data are the average of three experiments ± s.d. (N > 50, Pcp1-mCherry positive cells).

DOI: https://doi.org/10.7554/eLife.41418.009

The following source data and figure supplements are available for figure 4:

**Source data 1.** The source data to plot the graph in *Figure 4E*.
DOI: https://doi.org/10.7554/eLife.41418.016

**Figure supplement 1.** SAS-6-GFP localization within the fission yeast SPB revealed by structured illumination microscopy (SIM).
DOI: https://doi.org/10.7554/eLife.41418.010

**Figure supplement 1—source data 1.** The source data to plot the graph in *Figure 4—figure supplement 1C*.
DOI: https://doi.org/10.7554/eLife.41418.011

**Figure supplement 2.** *Drosophila* SAS-6 localizes to the fission yeast SPB through the C-terminal coiled-coil domain.

*Figure 4 continued*

DOI: https://doi.org/10.7554/eLife.41418.012

**Figure supplement 2—source data 1.** The source data to plot the graph in *Figure 4—figure supplement 2C*.

DOI: https://doi.org/10.7554/eLife.41418.013

**Figure supplement 3.** Localization of SAS-6 at the fission yeast SPB is cell-cycle regulated.

DOI: https://doi.org/10.7554/eLife.41418.014

**Figure supplement 3—source data 1.** The source data to plot the graphs in *Figure 4—figure supplement 3B and E*.

DOI: https://doi.org/10.7554/eLife.41418.015

recruiting centriole components. To test our prediction, we examined in animal cells whether pericentrin interacts with SAS-6 and helps to recruit it to the centriole.

The pericentrin family varies in protein length but contains the conserved PACT domain in the C-terminal region (*Figure 5A*). Since we observed that fission yeast Pcp1 interacts with SAS-6 through the PACT-domain containing region, we tested whether SAS-6 interacts with the PACT domain of *Drosophila* pericentrin ortholog, pericentrin-like protein (PLP). EGFP-tagged SAS-6 and HA-tagged PLP fragment containing the conserved PACT domain were co-expressed in *Drosophila melanogaster* tissue cultured cells (D.Mel cells). Consistent with the results obtained in fission yeast, SAS-6 interacts with the PACT domain (*Figure 5B*). To verify whether this interaction is direct, we validated this result in vitro, by performing in vitro binding assays using purified GST, GST-tagged SAS-6 and His-tagged PACT. GST-SAS-6 was specifically bound to His-PACT, indicating a direct interaction between these two proteins (*Figure 5C*).

## The conserved SAS-6-pericentrin interaction plays a role in centriole assembly

Pericentrin is highly expressed and correlates with the levels of centrosome aberrations in acute myeloid leukemia (AML) (*Krämer et al., 2005*; *Neben et al., 2004*). Moreover, overexpression of pericentrin in S-phase-arrested CHO cells induces the formation of numerous daughter centrioles (*Loncarek et al., 2008*). We thus wondered whether those effects were mediated by excessive SAS-6 recruitment since SAS-6 overexpression leads to supernumerary centriole formation (*Peel et al., 2007*; *Strnad et al., 2007*).

We first examined whether overexpression of EGFP-tagged PACT domain under the actin5C promoter has an effect on SAS-6 recruitment and centriole biogenesis. To enrich for mitotic cells, a stage where SAS-6 levels are homogeneous and highest (our own unpublished observations; see also recent publication [*Aydogan et al., 2018*]), D.Mel cells overexpressing either EGFP-PACT or EGFP were arrested in mitosis by a colchicine treatment (*Dobbelaere et al., 2008*; *Goshima et al., 2007*). Indeed, overexpressing PACT in cells led to increased recruitment of SAS-6 to centrioles, suggesting PACT recruits SAS-6 (*Figure 5D and E*). This is consistent with our findings in fission yeast (*Figure 4D*). Moreover, other centriole components may also be perturbed directly through PACT interaction or indirectly, given that SAS-6 is known to interact and recruit other centriole components, such as Bld10 and Ana2 (*Galletta et al., 2016*; *Jana et al., 2018*; *Lin et al., 2013*; *Stevens et al., 2010*). Since SAS-6 upregulation leads to centriole amplification (*Leidel et al., 2005*), we examined if PACT overexpression has the same phenotype. In the absence of electron microscopy, we confirmed the presence of supernumerary centrioles by staining with two combinations of reliable centriole markers (SAS-4 and Bld10, and SAS-4 and Ana1). Reflecting the increased recruitment of SAS-6 to centrioles, we observed a significant increase in the percentage of cells with more than four centrioles compared to the EGFP control (*Figure 5F and G*).

## SAS-6 is recruited to the centriole by two complementary pathways

In animal cells, it is known that STIL/Ana2/SAS-5 recruits SAS-6 to the centriole (*Arquint et al., 2012*; *Ohta et al., 2014*; *Moyer et al., 2015*). However, it has been observed that STIL depletion does not completely prevent HsSAS-6 recruitment, and contribution of STIL does not fully account for centrosomal targeting of HsSAS-6 during interphase (*Arquint et al., 2012*; *Keller et al., 2014*). These results suggest that other factor(s) may also recruit SAS-6 in animals. Since overexpression of PACT recruits more SAS-6, we examined if PLP has a role in recruiting SAS-6 to the centriole in

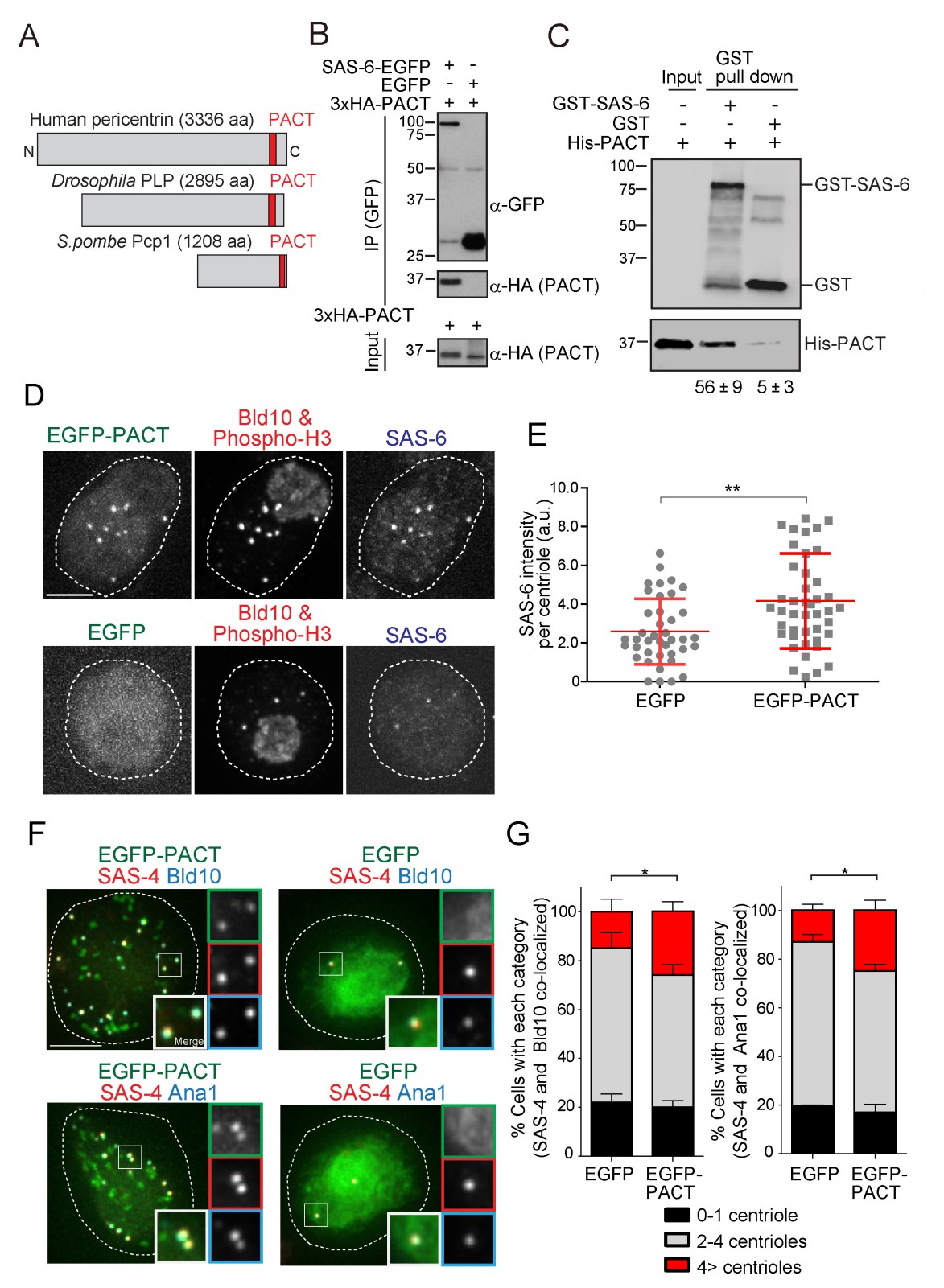

**Figure 5.** The *Drosophila* pericentrin (PLP) conserved domain –PACT- interacts with SAS-6, and its overexpression causes centriole amplification. (**A**) Schematic illustration of human pericentrin, *Drosophila* PLP and *S. pombe* Pcp1. (**B**) Physical interaction between SAS-6-EGFP and the conserved *Drosophila* PACT domain. Protein extract was prepared from *Drosophila* tissue culture cells (D.Mel cells) expressing HA-tagged PACT, and SAS-6-EGFP or EGFP. The GFP-tagged proteins were immunoprecipitated with anti-GFP antibody. Immunoprecipitates and inputs (20%) were analyzed by western

*Figure 5 continued on next page*

*Figure 5 continued*

blotting using the indicated antibodies. (C) Direct binding between SAS-6 and PACT. The in vitro binding assay was performed using purified GST or GST-fused SAS-6 and His-tagged PLP PACT. Note that we loaded 100% for each sample on each lane after pull-down to compare the efficiency of direct binding (bound vs input). Quantification of His-PACT bound to the GST-fusion protein is shown below the panel (data are the average of three experiments ± s.d) (D) Cells overexpressing EGFP-PACT or EGFP were arrested in mitosis by colchicine treatment for six hours and stained with the antibodies against Bld10 (centriole marker), phospho-H3 (mitotic marker) and SAS-6. Scale bar, 5 μm. (E) Quantification of SAS-6 intensity per centriole in cells overexpressing EGFP-PACT or EGFP arrested in mitosis. Means ± s.d. are shown in red (**p<0.001, Mann-Whitney U test). Results are representative of three independent experiments (N > 40 centrioles for each condition). (F) Cells overexpressing EGFP-PACT or EGFP were stained for two centriole markers (SAS-4 and Bld10, SAS-4 and Ana1) to count centriole number. Scale bar, 5 μm. (G) Quantification of centriole number per cell (N > 50, EGFP-positive cells). Data are the average of three experiments ± s.d. (*p<0.05, Mann-Whitney U test). Note that although control *Drosophila* tissue culture cells already show cells with underduplicated and over-duplicated centrioles as published before (*Bettencourt-Dias et al., 2005*), the expression of PACT leads to a significant amplification of centrioles.

DOI: https://doi.org/10.7554/eLife.41418.017

The following source data is available for figure 5:

**Source data 1.** The source data to plot the graphs in *Figure 5E and G*.
DOI: https://doi.org/10.7554/eLife.41418.018

addition to STIL/Ana2. We performed a single depletion of PLP and Ana2 and a co-depletion of both (Ana2 and PLP) in D.Mel cells. We used *mCherry* depletion as a negative control for this experiment. Firstly, we investigated if SAS-6 recruitment is impaired by *Ana2* and *PLP* RNAi. We focused on mitotic cells to compare all cells at the same cell cycle stage. The intensity of SAS-6 per centrosome was quantified in each RNAi condition. Upon depletion of Ana2 or PLP, SAS-6 intensity per pole was significantly reduced compared to control (*Figure 6A and B*). Moreover, double depletion of Ana2 and PLP caused an additive reduction of SAS-6 intensity. This result suggests SAS-6 is recruited by two complementary pathways, which depend on Ana2 and PLP. Western blotting analysis confirmed that the targeted proteins were efficiently depleted, while total SAS-6 protein levels were comparable in all conditions (*Figure 6C*). Considering that depletion of SAS-6 leads to a reduction in centriole number in both D.Mel cells and the fly (*Dobbelaere et al., 2008*; *Rodrigues-Martins et al., 2007*), we asked whether altered recruitment of SAS-6 by depleting Ana2 and PLP could also affect centriole biogenesis. We observed that co-depletion of Ana2 and PLP leads to stronger reduction in the number of centrioles as compared to the single depletions, indicating both pathways contribute to centriole biogenesis (*Figure 6D,E*, and *Figure 6—figure supplement 1*).

Furthermore, we asked if the centriole number reduction observed in PLP-depleted cells is attributable to PLP function or is an off-target effect, through a rescue experiment. We first depleted the endogenous PLP using another dsRNA against the 3'UTR region (*3'UTR PLP* RNAi) and expressed EGFP-tagged full-length PLP. Similar to *Figure 4D and E*, *3'UTR PLP* RNAi efficiently depleted endogenous PLP protein and caused a significant reduction in centriole number (*Figure 6—figure supplement 2*). Since we treated the cells with *3'UTR PLP* RNAi for two rounds aiming at a more efficient protein removal, the percentage of cells with reduced centriole number was slightly higher than in former *PLP* RNAi used for *Figure 6*. In contrast, when we expressed full-length PLP after PLP depletion, the centriole number defect was rescued, indicating the specific contribution of PLP for centriole formation (*Figure 6—figure supplement 2*). Moreover, we noticed that expression of full-length PLP in the presence of endogenous PLP induced centriole amplification similar to PACT over-expression (*Figure 6—figure supplement 2*).

### *Drosophila* pericentrin is required for SAS-6 recruitment and is important for centriole/basal body (BB) elongation in vivo

It is noteworthy that D.Mel cells are a sensitized system in which depletion of centriole components often shows an enhanced phenotype as compared to what is observed in the whole organism (the fly). For example, depletion of Bld10/CEP135 and CP110 by RNAi in D.Mel cells leads to a reduction in centriole number and length, respectively (*Carvalho-Santos et al., 2010*; *Delgehyr et al., 2012*), but the *Drosophila* mutants of these proteins only exhibit mild centriole defects (*Carvalho-Santos et al., 2010*; *Franz et al., 2013*; *Roque et al., 2012*). In previous studies using PLP-mutant flies, centriole number was not reduced, but centrioles were recently observed to be shorter in wing

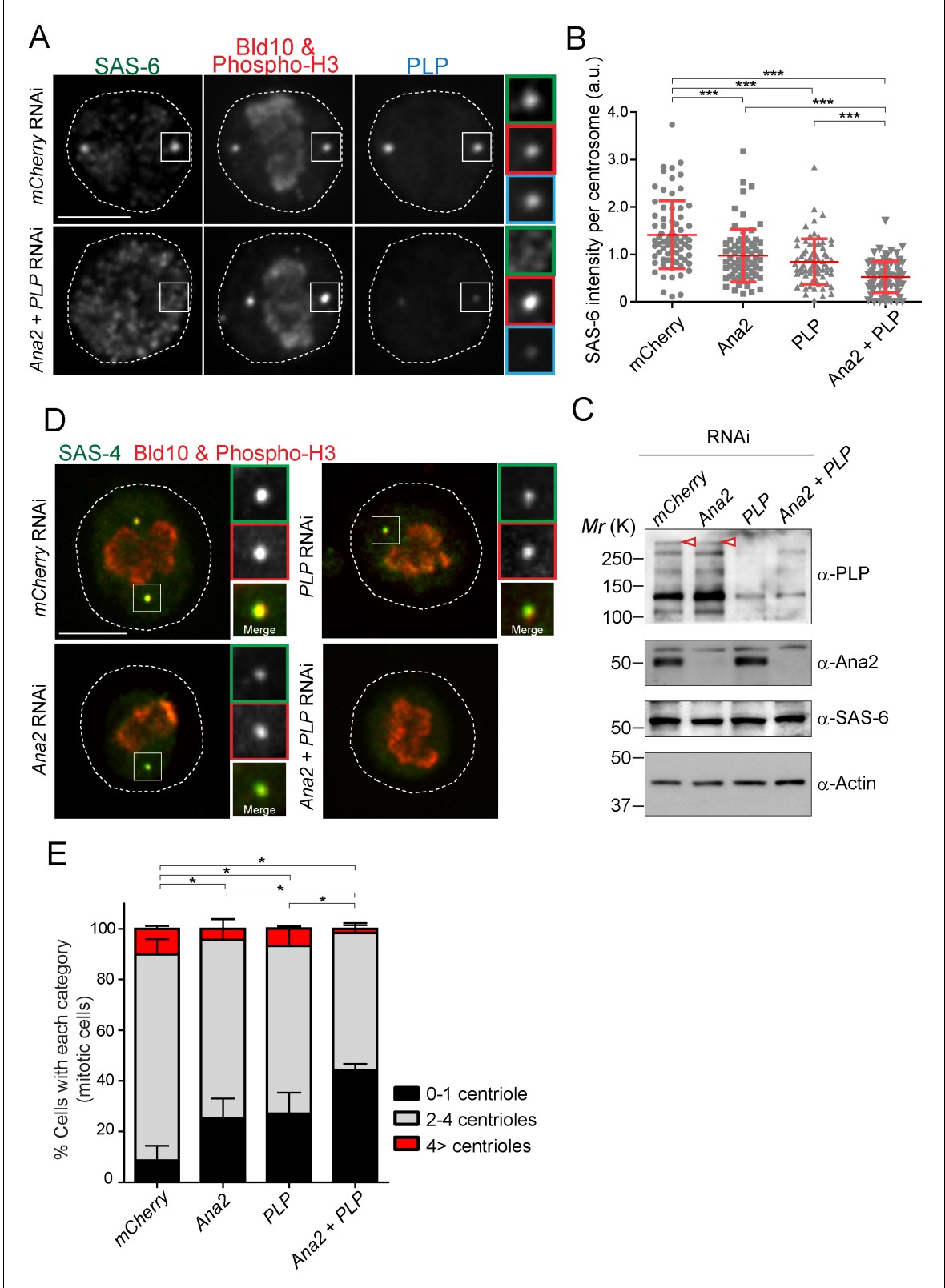

**Figure 6.** SAS-6-PACT complex formation complements the PLK4-STIL pathway in recruiting SAS-6 to the centrosome, and in promoting centriole biogenesis in tissue cultured cells. (**A**) Images of mitotic D.Mel cells after depletion of PLP and Ana2 by RNAi. D.Mel cells were depleted of PLP (*PLP* RNAi), Ana2 (*Ana2* RNAi), and were double-depleted with Ana2 and PLP (*Ana2 +PLP* RNAi) (*mCherry* RNAi was used as negative control) for three days. Cells were immunostained with anti-SAS-6, Bld10, phospho-H3 and PLP antibodies. Scale bar, 5 μm. (**B**) Quantification of the SAS-6 intensity per

*Figure 6 continued on next page*

*Figure 6 continued*
centrosome in the mitotic cells in the indicated RNAi conditions. Means ± s.d are shown in red (***p<0.001, Mann-Whitney U test). Results are representative of three independent experiments (N > 50 centrosomes for each condition). (C) Western blotting analysis of PLP, Ana2 and SAS-6 protein levels in the cells treated with the indicated dsRNAs using the antibodies against PLP, Ana2, SAS-6 and actin (loading control). Red arrowheads indicate expected bands of the longest PLP isoform. (D) Images of mitotic cells, after depletion of PLP and Ana2 by RNAi, used for centriole counting. Cells were immunostained with anti-Bld10 (red), phospho-H3 (red), and SAS-4 (another centriole marker, green) antibodies. Scale bar, 5 µm. (E) Quantification of centriole number per cell (N > 50). Data are the average of three experiments ± s.d (*p<0.05, Mann-Whitney U test performed for the 0–1 centriole category).

DOI: https://doi.org/10.7554/eLife.41418.019

The following source data and figure supplements are available for figure 6:

**Source data 1.** The source data to plot the graph in *Figure 6B and E*.
DOI: https://doi.org/10.7554/eLife.41418.024

**Figure supplement 1.** Counting of the centriole number in interphase cells.
DOI: https://doi.org/10.7554/eLife.41418.020

**Figure supplement 1—source data 1.** The source data to plot the graph in *Figure 6—figure supplement 1B*.
DOI: https://doi.org/10.7554/eLife.41418.021

**Figure supplement 2.** Expression of full-length PLP rescues centriole number defects in PLP-depleted cells.
DOI: https://doi.org/10.7554/eLife.41418.022

**Figure supplement 2—source data 1.** The source data to plot the graph in *Figure 6—figure supplement 2B*.
DOI: https://doi.org/10.7554/eLife.41418.023

disc cells (*Martinez-Campos et al., 2004*; *Roque et al., 2018*), suggesting an underappreciated generic role of PLP in centriole structure.

To further investigate the role of pericentrin in SAS-6 recruitment and centriole biogenesis we focused on fly spermatogenesis, as centrioles are converted to basal bodies to form cilia and elongate to more than 1 µm, becoming highly visible and easy to study. To investigate this, we depleted PLP during centriole assembly and basal body maturation in spermatocytes (*PLPRNAi*) using $Gal4^{Bam}$ (*Chen and McKearin, 2003*; see the timeline in *Figure 7A*). We studied its consequences in the localization of basal body components and basal body structure and, subsequently, in male fertility (*Figure 7*). The knockdown of PLP by RNAi did not affect centriole number in sperm cells (*Figure 7B and C*), suggesting that the STIL/Ana2 pathway recruits sufficient SAS-6 to initiate centriole biogenesis. However, PLP RNAi led to a decrease in centriole length, as observed with the PLP mutant in somatic cells (*Roque et al., 2018*), and to male infertility. Our laboratory has recently shown that SAS-6 is important for centriole elongation (*Jana et al., 2018*). We wondered whether PLP is necessary for recruiting a pool of SAS-6 that is needed for centriole elongation. Remarkably, PLP RNAi affected SAS-6 recruitment to the BBs (*Figure 7D and E*). Altogether, these results indicate that PLP is involved in sperm BB elongation by recruiting SAS-6, a phenotype that is further accentuated in tissue culture cells.

## The Calmodulin-PACT conserved interaction is likely to constrain the evolution of the PACT domain

We showed that the PACT domain has the capacity to interact with SAS-6 both in fission yeast and *Drosophila* cells, and the interaction between SAS-6 and pericentrin through PACT contributes to centriole assembly and elongation. It is possible that this interaction is ancestral, linking the two modules, the centriole and the PCM before the split between animals and fungi occurred one billion years ago. However, given the lack of centrioles and the divergence of the PCM in yeasts, we wondered why the PACT domain has retained the SAS-6 interaction surface. We hypothesized that other protein(s) that interact with the same region of the PACT domain as SAS-6, could be constraining the evolution of the interacting surface.

The PACT domain contains two highly conserved calmodulin (CaM)-binding domains (CBD1 and CBD2) (*Galletta et al., 2014*, *Figure 8A*). The interaction of pericentrin with calmodulin at the CaM-binding domain is conserved and is important for its function both in animals and yeasts. In *Drosophila*, this interaction controls the targeting of PLP to the centrosome and is, therefore, critical for its function (*Galletta et al., 2014*). The interaction of the budding yeast pericentrin, Spc110, with calmodulin, is required for the SPB to nucleate microtubules and to form the spindle (*Kilmartin and*

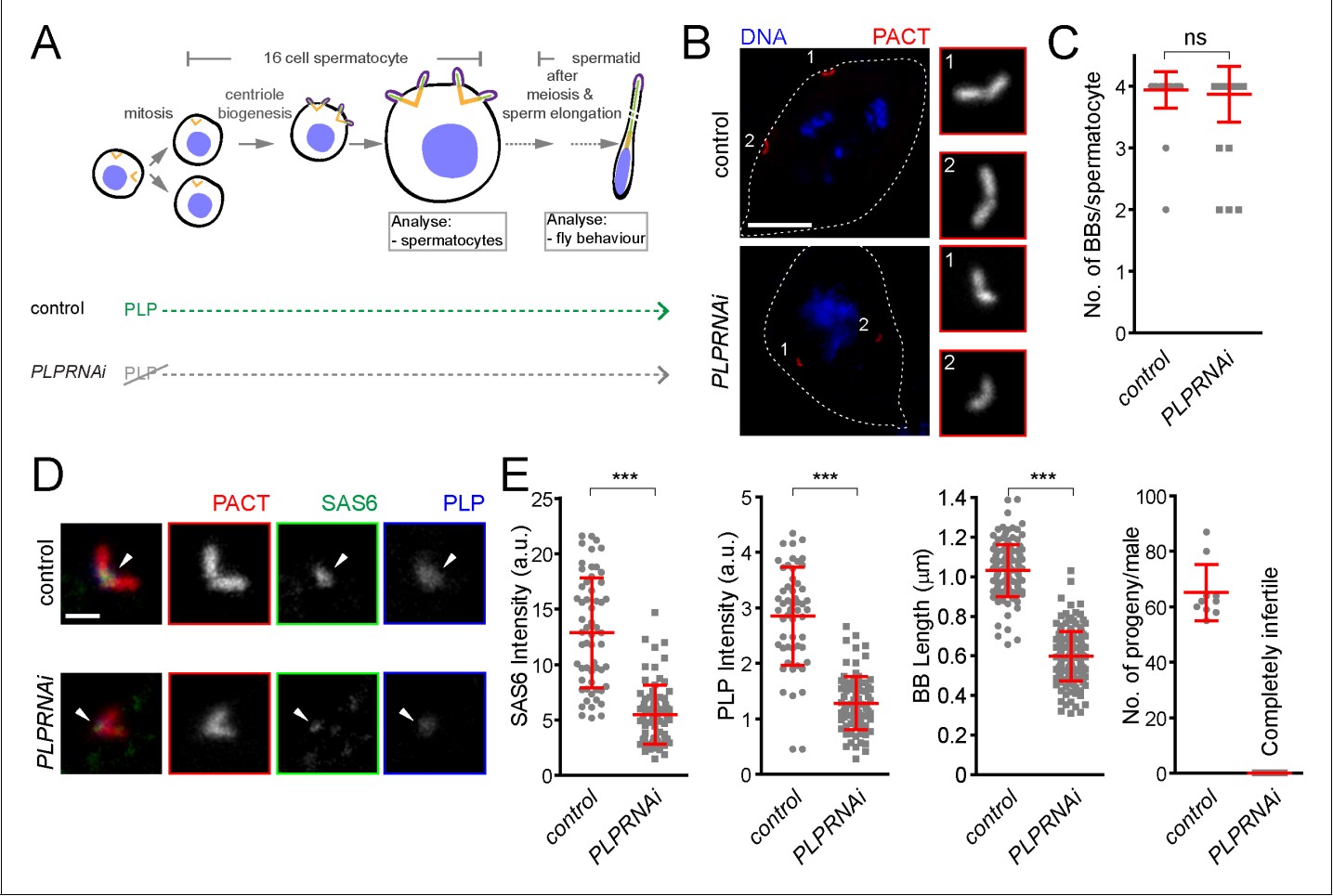

**Figure 7.** *Drosophila* pericentrin (PLP) is required for SAS-6 recruitment to the sperm centriole/basal body and for its elongation. (**A**) Schematic illustration of the experiments to deplete PLP during centriole biogenesis and elongation. (**B**) Representative images of mature spermatocytes in flies with different genotypes. PACT (red) is a commonly used marker for basal bodies (BB) and DAPI (blue) stains DNA. Insets show RFP::PACT close to the numbers (in gray scale). (**C**) Quantification of the number of BBs per cell in mature spermatocytes. (**D**) Representative images of mature spermatocyte BB in flies with different genotypes. RFP::PACT (red) marks BBs, Anti-SAS-6 (green) and Anti-PLP (blue) antibodies stain the proximal part of BB (arrowheads). (**E**) Quantification of SAS-6, PLP, BB length and the number of progeny in flies with different genotypes. We repeated all experiments three times. For SAS-6 and PLP intensities and BB length analysis, the number of BBs quantified for each genotype is N ≥ 108 (54 pairs of BBs) and N ≥ 128, respectively. The total number of males used for each histogram bar is N ≥ 10. Notably, given that moderate overexpression of PACT domain (using polyUbiquitin promoter) in the *plp* mutant fly fails to rescue the observed centriole as well as behavior defects of the mutant (***Martinez-Campos et al., 2004***), we used RFP::PACT to study the sperm basal bodies in the knockdown experiments. Scale bars in (**B**) and (**D**) represent 10 and 1 μm, respectively. Means ± s.d are shown in red (ns-not significant, ***p<0.001, Mann-Whitney U test).
DOI: https://doi.org/10.7554/eLife.41418.025

The following source data is available for figure 7:

**Source data 1.** The source data to plot the graph in ***Figure 7C and E***.
DOI: https://doi.org/10.7554/eLife.41418.026

***Goh, 1996***; ***Spang et al., 1996***; ***Stirling et al., 1994***; ***Stirling et al., 1996***). Therefore, we asked whether SAS-6 interacts with a conserved PLP segment, starting just before CBD1 and ending just after CBD2, hereafter called CBD (see ***Figure 8A*** in yellow). Similarly, as in ***Figure 5B***, we co-expressed EGFP-tagged SAS-6 and the HA-tagged CBD in D.Mel cells to examine their interaction. Indeed, similar to the PACT domain, CBD interacted with SAS-6-EGFP (***Figure 8B***). This result indicates that this conserved segment within the PACT domain is sufficient for interaction with SAS-6.

Next, we asked whether SAS-6 is present in the same complex with calmodulin. Intriguingly, we found that SAS-6-EGFP, CBD and calmodulin formed a complex (***Figure 8B***). Since SAS-6-EGFP and

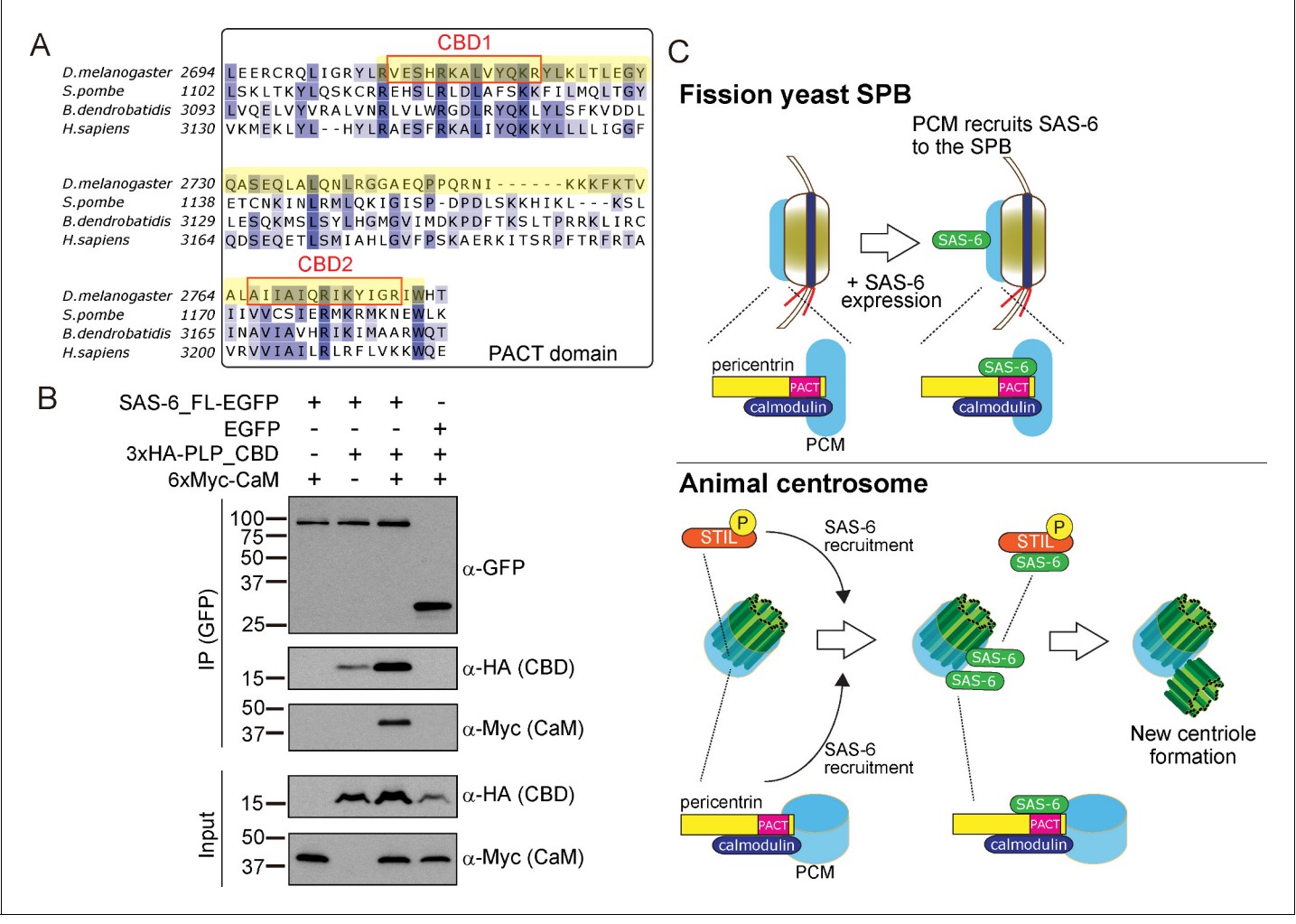

**Figure 8.** SAS-6 interacts with the calmodulin binding domain within PACT. (**A**) Graphical representations of the calmodulin (CaM)-binding domains (CBD1 and CBD2) within the multiple sequence alignment of the PACT domain of pericentrin proteins in the indicated species. CBD1 and CBD2 are marked with red squares. In yellow, minimal fragment containing both CBD1 and CBD2 (hereafter called CBD and used for subsequent experiments). The sequences of the PACT domain were aligned using Clustal Omega multiple sequence alignment tool and visually represented using Jalview software (*McWilliam et al., 2013*; *Waterhouse et al., 2009*). The alignments are color-coded in shades of blue for the percentage identity of amino acids; darkest blue (>80%), mid blue (>60%), light blue (>40%), white (<=40%). (**B**) Complex formation between SAS-6, the highly conserved CBD within PACT domain and calmodulin. Protein extract was prepared from the D.Mel cells expressing SAS6-EGFP or EGFP, HA-tagged CBD and Myc-tagged calmodulin. The GFP-tagged proteins were immunoprecipitated with anti-GFP antibody. Immunoprecipitates and inputs (20%) were analyzed by western blotting using the indicated antibodies. (**C**) Schematic representation of the ancestral role of the PCM in recruiting centriole proteins, centriole biogenesis and elongation. When SAS-6 was heterologously expressed in fission yeast cells, it localized to the SPB through interaction with Pcp1/pericentrin (upper panel). This revealed a novel interaction between SAS-6 and pericentrin in animals, which is important for centriole biogenesis (lower panel). It is likely that SAS-6 is recruited to the pre-existing centriole prior to new centriole formation by two complementary pathways: PLK4-STIL/Ana2 and pericentrin. Eventhough yeast and animals are separated by one billion years of evolution, the pericentrin/Pcp1-SAS-6 interaction surface has been conserved, likely because the binding of pericentrin to calmodulin constrained its evolution.

DOI: https://doi.org/10.7554/eLife.41418.027

calmodulin did not interact directly, we concluded that the complex forms through the CBD segment. Moreover, we observed that the CBD polypeptide was stabilised in the presence of SAS-6-EGFP, and this was more pronounced with the addition of calmodulin, suggesting that formation of this complex might stabilize pericentrin (*Figure 8B*). Since the binding of SAS-6 and calmodulin to the CBD was not competitive, it is likely that these two proteins interact with adjacent but distinct amino acids within the CBD. Further biochemical analysis will be necessary to estimate the exact stoichiometry of the protein complex.

Our result suggests that the fact that pericentrin function and stability has always relied on its interaction with calmodulin has constrained the evolution of the pericentrin's CBD's local structure, therefore retaining its affinity for SAS-6.

## Discussion

In this study, we examined the relationship between the PCM and centriole components, taking advantage of a heterologous system, the fission yeast, which does not have centrioles, and has lost the coding sequences for its components. Surprisingly, the *Drosophila* core centriole components SAS-6, Bld10 and SAS-4 localized to the fission yeast SPB. In particular, SAS-6 was specifically recruited to the SPB through its interaction with the conserved PCM component, Pcp1/pericentrin, via its PACT domain. Importantly, this interaction was also observed in animals (*Drosophila*). Further analysis revealed that pericentrin is required for both SAS-6 recruitment to centrioles in addition to the STIL/Ana2 pathway, and for proper centriole assembly, in particular, centriole elongation. It is estimated that animals and fungi diverged from their common ancestor about one billion years ago (*Douzery et al., 2004*; *Parfrey et al., 2011*). Our results reveal an evolutionarily conserved relationship between the centriole and the PCM, in which the PCM is needed for centriole component recruitment (*Figure 8C*). Therefore, the localization of new centrioles is likely to be dictated not just by positive regulatory feedbacks amongst centriole components, but also through the regulation of centriole component localization by the PCM.

### Implications of the pericentrin-SAS-6 interaction in centriole assembly and elongation

SAS-6 is a critical structural component of centrioles. It builds the cartwheel by self-assembly, which helps to define the centriole nine-fold symmetry (*Kitagawa et al., 2011*; *van Breugel et al., 2011*). It had been previously observed that the STIL/Ana2-dependent SAS-6 recruitment pathway did not account for all SAS-6 present at the centriole (*Arquint et al., 2012*; *Keller et al., 2014*), but alternative pathways for SAS-6 recruitment were not known. In this study, we serendipitously discovered that SAS-6 is also recruited to the centriole through another undescribed pathway mediated by pericentrin, though complementary to the previously characterized STIL/Ana2 pathway. Then, how and where does pericentrin interact with SAS-6? Given that in human cells, SAS-6 was previously seen to co-localize with pericentrin at the proximal end of the centriole where it accumulates from S phase (*Keller et al., 2014*), we suggest that pericentrin might recruit SAS-6 at that stage. However, in *Drosophila* tissue cultured cells, it has been shown that pericentrin forms a ring around the mother centriole and SAS-6 occupies the central region of the centriole, indicating that the two proteins may not show overlapping localization (*Dzhindzhev et al., 2014*; *Mennella et al., 2012*). We propose that the pericentrin-SAS-6 interaction might be transient and highly dynamic and therefore co-localization is more difficult to observe.

Previous biophysical studies have shown that self-interaction between the head group of SAS-6 is relatively weak (characterized by micro-molar range dissociation constant), which suggested the existence of other factors that contribute to the nine-fold symmetry of the centriole (*Cottee et al., 2011*). Indeed, further work revealed that STIL/Ana2/SAS-5 forms higher-order assemblies and it is likely to concentrate SAS-6 molecules and contribute to cartwheel formation in *C. elegans* and *Drosophila* (*Cottee et al., 2015*; *Rogala et al., 2015*). We propose that pericentrin might also concentrate SAS-6 around the PCM-bearing mother centriole, and thus contribute to centriole formation, in particular, its elongation, in addition to the STIL/Ana2 pathway. This role would be similar to the recently discussed role of the PCM in concentrating MT nucleators (*Woodruff et al., 2017*). Our data also provide a framework to understand why pericentrin overexpression leads to centrosome amplification (*Krämer et al., 2005*; *Loncarek et al., 2008*; *Neben et al., 2004*).

How are pericentrin and SAS-6 implicated in centriole elongation? Previous studies showed that SAS-6 is important for centriole maintenance (*Izquierdo et al., 2014*), and expression of a symmetry-changing SAS-6 mutant leads to reduced centriole length in human cells (*Hilbert et al., 2016*). That work suggests that intactness of the SAS-6 protein is indispensable for centriole stability and length control. We recently reported that SAS-6 is dynamically recruited to the spermatocyte BBs to promote elongation (*Jana et al., 2018*). Here, our data suggest that pericentrin regulates centriole length and stability by recruiting a dynamic pool of SAS-6 to the centriole/BB.

## Calmodulin-binding to pericentrin may have constrained the evolution of the pericentrin-SAS-6 interaction surface

We proposed that the binding of molecules such as calmodulin, to the PACT domain of pericentrin, might constrain PACT evolution. This is also the domain that interacts with SAS-6. Calmodulin binding is important for pericentrin centriole-targeting, and also for its function and protein stability (*Galletta et al., 2014*). Moreover, the fission yeast *pcp1-14* mutant, which harbors a point mutation in the residue adjacent to CBD2, also exhibits reduced protein stability (*Tang et al., 2014*), implying a conserved role of calmodulin binding in stabilizing Pcp1/PLP. Indeed, we observed that calmodulin binding stabilized PLP (*Figure 8B*). Previous studies in budding yeast suggested that calmodulin binding to the C-terminal region of Spc110 regulates the interaction between Spc110 and a critical inner plaque component, Spc29 (*Elliott et al., 1999*). Perhaps calmodulin binding modulates the conformation of pericentrin into an 'active form' which is stable and capable of interacting with the centriole and SPB component(s), similarly to the regulation of calmodulin-dependent kinases by calmodulin (*Crivici and Ikura, 1995*). Intriguingly, it is reported that a single amino acid deletion of the conserved lysine residue (3154K) mapped in CBD1 of the human pericentrin gene is associated with a genetic syndrome microcephalic primordial dwarfism type II (MOPDII), characterized with severe developmental anomalies (*Kantaputra et al., 2011*). Although the contribution of this mutation at the cellular level is not known, it is possible that this disease phenotype is caused by abnormal centriole formation due to failure to recruit SAS-6, as we observed in this study. To support this possibility, it is also known that mutation in the evolutionary-conserved PISA (present in SAS-6) motif of the SAS-6 gene (*Leidel et al., 2005*) that affects its function in centriole formation, is associated with autosomal recessive primary microcephaly (*Khan et al., 2014*).

## Implications for the study of cellular evolution and future application

Although fission yeast and other fungi lost centrioles, while acquiring the centrosome-equivalent SPBs, our study indicates that fission yeast SPBs still retain part of the ancestral PCM structure. Centrioles have a critical role in motility as basal bodies of cilia/flagella and in microtubule nucleation by recruiting the PCM. After centriole loss upon the lack of requirement for flagella-based movement, the necessity to maintain microtubule nucleation might have constrained the evolution of the PCM structure. Indeed, pericentrin orthologs in fungi, play essential roles in microtubule organization, mitotic spindle assembly, and cell cycle regulation similarly as in animals, and their function is dependent on calmodulin-binding (*Geiser et al., 1993*; *Kilmartin et al., 1993*; *Flory et al., 2002*; *Fong et al., 2010*; *Chen et al., 2012*). We suggest that the critical role of the calmodulin-pericentrin interaction for proper microtubule nucleation has been retained in evolution, and constrained the evolution of the pericentrin structure even after other interactors, such as SAS-6, were lost upon centriole loss. Our results support the hypothesis that the yeast SPB evolved from an ancestral centriolar centrosome by a step-wise remodeling process: an SPB precursor appeared on the centrosome, interacting with the PCM and replacing the role of the centriole as a PCM-recruiting entity (*Ito and Bettencourt-Dias, 2018*; *McLaughlin et al., 2015*).

It has been shown that many of the yeast genes can be substituted by human orthologs (e.g. rescuing the fission yeast *cdc2* mutant with the human CDK1 gene), indicating that critical ancestral functions have been conserved across a billion years (*Lee and Nurse, 1987*; *Osborn and Miller, 2007*; *Kachroo et al., 2015*). The present study demonstrated that critical functional modules can retain not only the same function, but also interaction capacities even when a binding partner is completely lost. The approach to heterologously expressing evolutionary-lost components of organelles in diverse organisms/cells could be useful to identify such novel interactions and important conserved interaction domains and divergent orthologue proteins across species.

Finally, since fission yeast SPB recruits centriole proteins, it might be feasible to use fission yeast to assemble multiple centriole components at the SPBs. This synthetic biological approach could be useful to study the process of centriole biogenesis, such as the interaction between components and the order of recruitment, which will ultimately lead to the successful reconstitution of the evolutionary-lost centriole structure in fission yeast.

# Materials and methods

## Key resources table

| Reagent type (species) or resource | Designation | Source or reference | Identifiers | Additional information |
|---|---|---|---|---|
| Strain (*Schizosaccharomyces pombe*) | h- leu1-32 ura4-D18 lys1::P$^{atb2}$-DmSAS-6-GFP-FLAG-6xHis-ura4 + sid4-tdTomato-natMX6 | This paper | DI 456 | *Figure 2A,B, Figure 2—figure supplement 1A,B, Figure 4—figure supplement 3C,D* |
| Strain (*Schizosaccharomyces pombe*) | h- leu1-32 ura4-D18 lys1::P$^{atb2}$-DmBld10-GFP-FLAG-6xHis-ura4 + sid4-mRFP-natMX6 | This paper | DI 190 | *Figure 2A,B* |
| Strain (*Schizosaccharomyces pombe*) | h- leu1-32 ura4-D18 lys1::P$^{atb2}$-6xHis-FLAG-YFP-DmSAS-4-ura4 + sid4-mRFP-natMX6 | This paper | DI 202 | *Figure 2A,B* |
| Strain (*Schizosaccharomyces pombe*) | h- ura4-D18 lys1::P$^{nmt1}$-6xHis-FLAG-YFP-DmAna2-ura4 + sid4-tdtomato-natMX6 | This paper | DI 655 | *Figure 2A,B* |
| Strain (*Schizosaccharomyces pombe*) | h + ura4-D18 leu1::P$^{nmt41}$-6xHis-FLAG-GFP-DmPlk4-ura4 + sid4-tdtomato-natMX6 | This paper | DI 665 | *Figure 2A,B* |
| Strain (*Schizosaccharomyces pombe*) | h- leu1-32 ura4-D18 lys1::P$^{atb2}$-DmSAS-6-GFP-FLAG-6xHis-ura4 + sid4-tdtomato-natMX6 pcp1-HA-hphMX6 cut7-446 | This paper | DI 492 | *Figure 2C, Figure 3* |
| Strain (*Schizosaccharomyces pombe*) | h + leu1-32 ura4-D18 lys1::P$^{atb2}$-DmBld10-GFP-FLAG-6xHis-ura4 + sid4-tdtomato-natMX6 pcp1-HA-hphMX6 cut7-446 | This paper | DI 706 | *Figure 2C, Figure 3* |
| Strain (*Schizosaccharomyces pombe*) | h + leu1-32 ura4-D18 lys1::P$^{atb2}$-6xHis-FLAG-YFP-DmSAS-4-ura4 + sid4-tdtomato-natMX6 pcp1-HA-hphMX6 cut7-446 | This paper | DI 709 | *Figure 2C, Figure 3* |
| Strain (*Schizosaccharomyces pombe*) | h- leu1-32 ura4-D18 lys1::P$^{atb2}$-GFP-FLAG-6xHis-ura4 + sid4-tdtomato-natMX6 pcp1-HA-hphMX6 cut7-446 | This paper | DI 486 | *Figure 2C* |

*Continued on next page*

*Continued*

| Reagent type (species) or resource | Designation | Source or reference | Identifiers | Additional information |
|---|---|---|---|---|
| Strain (*Schizosaccharomyces pombe*) | h- leu1-32 ura4-D18 | This paper | DI 7 | *Figure 2C, Figure 3, Figure 2— figure supplement 1A,B, Figure 4— figure supplement 2A–C, Figure 4— figure supplement 3A,B* |
| Strain (*Schizosaccharomyces pombe*) | h- leu1-32 ura4-D18 lys1::P$^{atb2}$-DmSAS-6-GFP-FLAG-6xHis-ura4 + sid4-tdtomato-natMX6 pcp1-14-HA-hphMX6 cut7-446 | This paper | DI 631 | *Figure 3* |
| Strain (*Schizosaccharomyces pombe*) | h + leu1-32 ura4-D18 lys1::P$^{atb2}$-DmBld10-GFP-FLAG-6xHis-ura4 + sid4-tdtomato-natMX6 pcp1-14-HA-hphMX6 cut7-446 | This paper | DI 710 | *Figure 3* |
| Strain (*Schizosaccharomyces pombe*) | h- leu1-32 ura4-D18 lys1::P$^{atb2}$-6xHis-FLAG-YFP-DmSAS-4-ura4 + sid4-tdtomato-natMX6 pcp1-14-HA-hphMX6 cut7-446 | This paper | DI 719 | *Figure 3* |
| Strain (*Schizosaccharomyces pombe*) | h + leu1-32 ura4-D18 lys1::P$^{atb2}$-DmSAS-6-GFP-FLAG-6xHis-ura4+ | This paper | DI 105 | *Figure 4B* |
| Strain (*Schizosaccharomyces pombe*) | h- leu1-32 ura4-D18 lys1::P$^{atb2}$-DmSAS-6-GFP-FLAG-6xHis-ura4 + sfi1-CFP-natMX6 | This paper | DI 636 | *Figure 4D,E* |
| Strain (*Schizosaccharomyces pombe*) | h- leu1-32 ura4-D18 lys1::P$^{atb2}$-GFP-FLAG-6xHis-ura4 + sfi1-CFP-natMX6 | This paper | DI 638 | *Figure 4D,E* |
| Strain (*Schizosaccharomyces pombe*) | h- leu1-32 ura4-D18 lys1::P$^{atb2}$-DmSAS-6-GFP-FLAG-6xHis-ura4 + sid4-tdTomato-natMX6 | This paper | DI 456 | *Figure 2— figure supplement 1A,B* |
| Strain (*Schizosaccharomyces pombe*) | h- leu1-32 ura4-D18 lys1::P$^{atb2}$-DmBld10-GFP-FLAG-6xHis-ura4 + sid4-mRFP-natMX6 | This paper | DI 190 | *Figure 2— figure supplement 1A,B* |

*Continued on next page*

*Continued*

| Reagent type (species) or resource | Designation | Source or reference | Identifiers | Additional information |
|---|---|---|---|---|
| Strain (*Schizosaccharomyces pombe*) | h- leu1-32 ura4-D18 lys1::P$^{atb2}$-6xHis-FLAG-YFP-DmSAS-4 -ura4 + sid4-mRFP-natMX6 | This paper | DI 202 | *Figure 2—figure supplement 1A,B* |
| Strain (*Schizosaccharomyces pombe*) | h- ura4-D18 lys1::P$^{nmt1}$-6xHis-FLAG -YFP-DmAna2-ura4 + sid4-tdtomato-natMX6 | This paper | DI 655 | *Figure 2—*<br><br>*figure supplement 1A,B* |
| Strain (*Schizosaccharomyces pombe*) | h- leu1-32 ura4-D18 | This paper | DI 7 | *Figure 2—figure supplement 1A,B* |
| Strain (*Schizosaccharomyces pombe*) | h- leu1-32 ura4-D18 pcp1-GFP-kanMX6 sid4-tdtomato-natMX6 | This paper | DI 547 | *Figure 4—figure supplement 1A,B* |
| Strain (*Schizosaccharomyces pombe*) | h- leu1-32 ura4-D18 arg1:: P$^{nmt1}$-DmSAS-6-GFP-FLAG-6xHis-kanMX6 sid4-tdtomato -natMX6 | This paper | DI 646 | *Figure 4—figure supplement 1A,B* |
| Strain (*Schizosaccharomyces pombe*) | h- leu1-32 ura4-D18 arg1::P$^{nmt1}$-DmSAS-6-GFP-FLAG-6xHis-kanMX6 pcp1-td Tomato-natMX6 | This paper | DI 721 | *Figure 4—*<br><br>*figure supplement 1A,B* |
| Strain (*Schizosaccharomyces pombe*) | h- leu1-32 ura4-D18 arg1::P$^{nmt1}$-DmSAS-6-GFP-FLAG-6xHis-kanMX6 sid4-tdtomato-natMX6 | This paper | DI 646 | *Figure 4—figure supplement 2A–C* |
| Strain (*Schizosaccharomyces pombe*) | h- leu1-32 ura4-D18 arg1::P$^{nmt1}$-DmSAS-6 (Reg7, 1–176 aa)-GFP-FLAG -6xHis-kanMX6 sid4-tdTomato-natMX6 | This paper | DI 671 | *Figure 4—figure supplement 2A–C* |
| Strain (*Schizosaccharomyces pombe*) | h- leu1-32 ura4-D18 arg1::P$^{nmt1}$-DmSAS-6 (Reg6, 177–472 aa)-GFP-FLAG-6xHis-kanMX6 sid4-tdtomato-natMX6 | This paper | DI 648 | *Figure 4—figure supplement 2A–C* |
| Strain (*Schizosaccharomyces pombe*) | h- leu1-32 ura4-D18 lys1::P$^{atb2}$-DmSAS-6-GFP-FLAG-6xHis-ura4 + sid4-td Tomato-natMX6 pcp1-HA-hphMX6 cdc25-22 | This paper | DI 454 | *Figure 4—figure supplement 3A,B* |
| Recombinant DNA reagent (plasmid) | pLYS1U-GFH21c-DmSAS-6 | This paper | | Expression of DmSAS-6-GFP in *S. pombe* (*atb2* promoter) |

*Continued on next page*

*Continued*

| Reagent type (species) or resource | Designation | Source or reference | Identifiers | Additional information |
|---|---|---|---|---|
| Recombinant DNA reagent (plasmid) | pLYS1U-GFH21c-DmBLD10 | This paper | | Expression of DmBld10-GFP in *S. pombe* (*atb2* promoter) |
| Recombinant DNA reagent (plasmid) | pLYS1U-HFY21c-DmSAS-4 | This paper | | Expression of YFP-DmSAS-4 in *S. pombe* (*atb2* promoter) |
| Recombinant DNA reagent (plasmid) | pLYS1U-HFY1c-DmAna2 | This paper | | Expression of YFP-DmAna2 in *S. pombe* (*nmt1* promoter) |
| Recombinant DNA reagent (plasmid) | pDUAL2-HFG1c-DmPlk4 | This paper | | Expression of GFP-DmPlk4 in *S. pombe* (*nmt1* promoter) |
| Recombinant DNA reagent (plasmid) | pARG1-GFH1c-DmSAS-6 | This paper | | Expression of DmSAS-6-GFP in *S. pombe* (*nmt1* promoter) |
| Recombinant DNA reagent (plasmid) | pARG1-GFH1c-DmSAS-6 (Reg7, 1–176 aa) | This paper | | Expression of DmSAS-6-GFP (Reg7, 1–176 aa) in *S. pombe* (*nmt1* promoter) |
| Recombinant DNA reagent (plasmid) | pARG1-GFH1c-DmSAS-6(Reg6, 177–472 aa) | This paper | | Expression of DmSAS-6-GFP (Reg6, 177–472 aa) in *S. pombe* (*nmt1* promoter) |
| Recombinant DNA reagent (plasmid) | pREP41-pcp1-mcherry | This paper | | Expression of Pcp1-mCherry (full length) in *S. pombe* (*nmt41* promoter) |
| Recombinant DNA reagent (plasmid) | pREP41-pcp1 (N 1–419)-mcherry | This paper | | Expression of Pcp1-mCherry (N 1–419) in *S. pombe* (*nmt41* promoter) |
| Recombinant DNA reagent (plasmid) | pREP41-pcp1 (M 420–782)-mcherry | This paper | | Expression of Pcp1-mCherry (M 420–782) in *S. pombe* (*nmt41* promoter) |
| Recombinant DNA reagent (plasmid) | pREP41-pcp1 (C 783–1208)-mcherry | This paper | | Expression of Pcp1-mCherry (C 783–1208) in *S. pombe* (*nmt41* promoter) |
| Recombinant DNA reagent (plasmid) | pAWG-DmSAS-6 | This paper | | Expression of DmSAS-6-EGFP in D.Mel cells (*actin5c* promoter) |
| Recombinant DNA reagent (plasmid) | pAGW | This paper | DGRC:1071 | Expression of EGFP in D.Mel cells (*actin5c* promoter) |
| Recombinant DNA reagent (plasmid) | pAHW-DmPLP_PACT | This paper | | Expression of 3xHA-DmPLP_PACT in D.Mel cells (*actin5c* promoter) |

*Continued on next page*

*Continued*

| Reagent type (species) or resource | Designation | Source or reference | Identifiers | Additional information |
|---|---|---|---|---|
| Recombinant DNA reagent (plasmid) | pAHW-DmPLP_CBD | This paper | | Expression of 3xHA-DmPLP_CBD in D.Mel cells (*actin5c* promoter) |
| Recombinant DNA reagent (plasmid) | pAMW-DmCaM | This paper | | Expression of 6xMyc-DmCaM in D.Mel cells (*actin5c* promoter) |
| Recombinant DNA reagent (plasmid) | pAGW-DmPLP_PACT | This paper | | Expression of EGFP-DmPLP_PACT in D.Mel cells (*actin5c* promoter) |
| Recombinant DNA reagent (plasmid) | pAGW-DmPLP_FL | This paper | | Expression of EGFP-DmPLP_FL in D.Mel cells (*actin5c* promoter) |
| Recombinant DNA reagent (plasmid) | pDEST15-DmSAS-6 | This paper | | Expression of GST-DmSAS-6 (full length) in *E. coli* |
| Recombinant DNA reagent (plasmid) | pGEX6p-1 | GE Healthcare | | Expression of GST in *E. coli* |
| Recombinant DNA reagent (plasmid) | pET30b-DmPLP_PACT | This paper | | Expression of 6xHis-DmPLP_PACT in *E. coli* |
| Cell line (*Drosophila melanogaster*) | D.Mel cells | Thermo Fisher Scientific | ATCC Cat# CRL-1963, RRID: CVCL_Z232 | *Drosophila* cultured cells |
| Sequence-based reagent | PLP-Forward primer (dsRNA synthesis (PLP)) | This paper | | TAATACGACT CACTATAGGGA GAGGAGCGCC TAAAGAACAGTG |
| Sequence-based reagent | PLP-Reverse primer (dsRNA synthesis (PLP)) | This paper | | TAATACGAC TCACTATAGGGA GACTGATCGA GCTGTTTGTGGA |
| Sequence-based reagent | Ana2-Forward primer (dsRNA synthesis (Ana2)) | This paper | | GAATTAATACGACTCA CTATAGGGA GAATGTTTGTTC CCGAAACGGAGG |
| Sequence-based reagent | Ana2-Reverse primer (dsRNA synthesis (Ana2)) | This paper | | GAATTAATACGACTC ACTATAGGGA GACAGAGCC GCCAGATCACTCTTA |
| Sequence-based reagent | mCherry-Forward primer (dsRNA synthesis (mCherry)) | This paper | | ATAATACGA CTCACTAT AGGGATGGTG AGCAAGGG |
| Sequence-based reagent | mCherry-Reverse primer (dsRNA synthesis (mCherry)) | This paper | | ATAATACGA CTCACTATA GGGGTTGAC GTTGTAGG |
| Sequence-based reagent | plp_3UTR_Forward primer (dsRNA synthesis (PLP_3'UTR)) | This paper | | TAATACGACT CACTATAGGGAG AGCCCAGGA TAGCAGAGTTGAG |

*Continued on next page*

Continued

| Reagent type (species) or resource | Designation | Source or reference | Identifiers | Additional information |
|---|---|---|---|---|
| Sequence-based reagent | plp_3UTR_Reverse primer (dsRNA synthesis (PLP_3'UTR)) | This paper | | TAATACGACT CACTATAGGGAGA CGAATGTGAAATAAAT TTGGTTTAA |
| Strain (*Drosophila melanogaster*) | w[1118]; Ubq-RFP::PACT;+ | *Carvalho-Santos et al., 2010* | | |
| Strain (*Drosophila melanogaster*) | w[1118]; +; bam[Gal4] | *Chen and McKearin, 2003* | | |
| Strain (*Drosophila melanogaster*) | yv; +; UAS-mCherry RNAi | *Perkins et al., 2015* | | |
| Strain (*Drosophila melanogaster*) | w[1118]; UAS-PLPRNAi; + | *Dietzl et al., 2007* | | |
| Strain (*Drosophila melanogaster*) | w[1118]; Ubq-RFP::PACT/+; UAS-mCherryRNAi/ bam[Gal4] | This paper | | |
| Strain (*Drosophila melanogaster*) | w[1118]; Ubq-RFP::PACT/ UAS-PLPRNAi; bam[Gal4]/+ | This paper | | |
| Antibody | anti-GFP (rabbit polyclonal) | Abcam | Abcam Cat# ab290, RRID:AB_303395 | WB 1:1000 |
| Antibody | anti-RFP (rat monoclonal) | Chromotek | RRID:AB_2336064 | WB 1:1000 |
| Antibody | anti-HA (rat monoclonal) | Roche | Roche Cat# 1 1867431001, RRID: AB_390919 | WB 1:1000 |
| Antibody | anti-Cdc2 PSTAIRE (rabbit polyclonal) | Santa Cruz Biotechnology | Cat# sc-53, RRID: AB_2074908 | WB 1:2000 |
| Antibody | anti-Drosophila SAS-6 (rabbit polyclonal) | Gift from J Gopalakrishnan | | WB 1:500 |
| Antibody | anti-Drosophila SAS-6 (rat polyclonal) | Gift from N Dzhindzhev and D Glover | | IF 1:500 |
| Antibody | anti-Drosophila Ana2 (rat polyclonal) | Gift from N Dzhindzhev and D Glover | | WB 1:4000 |
| Antibody | anti-phospho Histone H3 (Ser10) (rabbit polyclonal) | Millipore | Millipore Cat# 06–570, RRID:AB_310177 | IF 1:2000 |
| Antibody | anti-Drosophila Bld10 (rabbit polyclonal) | Gift from T Megraw | | IF 1:5000 |
| Antibody | anti-Drosophila PLP (guinea pig polyclonal) | Gift from G Rogers | | WB 1:1000 |
| Antibody | anti-Drosophila PLP (chicken polyclonal) | *Bettencourt-Dias et al., 2005* | | IF 1:500 |

*Continued*

| Reagent type (species) or resource | Designation | Source or reference | Identifiers | Additional information |
|---|---|---|---|---|
| Antibody | anti-Actin (rabbit polyclonal) | Sigma-Aldrich | Sigma-Aldrich Cat# A2066, RRID: AB_476693 | WB 1:2000 |
| Antibody | anti-GST (mouse monoclonal) | Cell Signaling Technology | Cell Signaling Technology Cat# 3513, RRID: AB_1642209 | WB 1:1000 |
| Antibody | anti-His-tag (mouse monoclonal) | Millipore | Millipore Cat# 70796–3, RRID: AB_11213479 | WB 1:1000 |
| Antibody | anti-Myc (9E10) (mouse monoclonal) | Santa Cruz Bio technology | Santa Cruz Bio technology Cat# sc-40, RRID: AB_627268 | WB 1:1000 |
| Antibody | anti-Rat IgG (secondary, DyLight 488, Donkey) | Bethyl Laboratories | | IF 1:100 |
| Antibody | anti-Rabbit IgG (secondary, Rhodamine-Red, Donkey) | Jackson ImmunoResearch | | IF 1:100 |
| Antibody | anti-Rabbit IgG (secondary, Cy5, Donkey) | Jackson ImmunoResearch | | IF 1:100 |
| Antibody | anti-Chicken IgY (secondary, Cy5, Donkey) | Jackson ImmunoResearch | | IF 1:100 |
| Antibody | anti-Rat IgG (secondary, Cy5, Donkey) | Jackson ImmunoResearch | | IF 1:100 |
| Antibody | anti-Mouse IgG (secondary, HRP-conjugated, Donkey) | Jackson Immuno Research | | WB 1:5000 |
| Antibody | anti-Rabbit IgG (secondary, HRP-conjugated, Donkey) | Jackson Immuno Research | | WB 1:5000 |
| Antibody | anti-Guinea pig IgG (secondary, HRP-conjugated, Donkey) | Jackson ImmunoResearch | | WB 1:5000 |
| antibody | anti-Rat IgG (secondary, HRP-conjugated, Goat) | Bethyl Laboratories | | WB 1:5000 |
| antibody | anti-Rat IgG (secondary, IRDye 800CW, Goat) | LI-COR | | WB 1:10000 |
| antibody | anti-Mouse IgG (secondary, IRDye 680CW, Goat) | LI-COR | | WB 1:10000 |

### Fission yeast strains and culture

The *S. pombe* strains used in this study are listed in Key Resources Table. The strains were grown in yeast extract with supplements media (YE5S) or synthetic Edinburgh minimal media (EMM) in which ammonium chloride is replaced with glutamic acid (also called as PMG) with appropriate nutrient supplements as previously described (*Moreno et al., 1991*; *Petersen and Russell, 2016*).

### Plasmid DNA constructions

Integration and expression vectors for *S. pombe* and *Drosophila* cells used in this study were constructed using the Gateway system (Invitrogen). All cDNA encoding *Drosophila* centriole components (SAS-6, Bld10, SAS-4, Ana2 and Plk4), PACT domain and calmodulin-binding domain of PACT (CBD) were amplified by PCR and cloned into pDONR221 vector.

To create integration plasmid for *S. pombe*, the pLYS1U-GFH21c (atb2 promoter, C-terminal GFP tag), pLYS1U-HFY1c (atb2 promoter, N-terminal YFP tag), pLYS1U-HFY1c (nmt1 promoter, N-terminal YFP tag), and pDUAL2-HFG1c (nmt1 promoter, N-terminal GFP tag) destination vectors (RIKEN Bioresource Center, Japan) were used. To express proteins in D.Mel cells, we used the destination vectors (*Drosophila* Genomics Resource Center, DGRC) containing the actin promoter termed: pAWG for the C-terminal EGFP tag, pAGW for N-terminal EGFP tag, pAHW for the N-terminal 3xHA tag, and pAMW for the N-terminal 6xMyc tag.

The plasmids for overexpression of fission yeast full length and truncated Pcp1-mCherry were constructed as follows. Each region of the *pcp1+* gene was amplified by PCR and inserted into the SalI-NotI site of the expression plasmids pREP41 with mCherry at the carboxyl terminus in which gene expression is controlled under nmt1 promoter (*Maundrell, 1990*).

For recombinant protein expression, we used pGEX6p-1 (GE Healthcare) for GST and the Gateway pDEST15 (N-terminal GST, Thermo Fisher Scientific) destination vector for GST-DmSAS-6. To express 6xHis-PACT, the region of the PACT was amplified by PCR and inserted into the SalI-NotI site of the expression plasmid pET30-b (N-terminal 6xHis, Novagen). The plasmids used in this study are listed in Key Resources Table.

### Gene targeting and strain construction in fission yeast

To generate the fission yeast strains expressing *Drosophila* centriole proteins SAS-6, Bld10 and SAS-4, the integration plasmids were linearized by digesting with NotI and integrated into each chromosomal locus. The integration into the targeted locus was verified by PCR.

### Expression of *Drosophila* centriole proteins in fission yeast

We initially used the strong inducible *nmt1* promoter (*Maundrell, 1990*) to screen the localization of five *Drosophila* centriole proteins (SAS-6, Bld10, SAS-4, Ana2 and Plk4) in fission yeast. Gene expression under the *nmt1* promoter was induced by removing thiamine from the culture media. We observed that while Ana2 and Plk4 did not localize to the SPB (*Figure 2A*), the other three localized there (data not shown). Subsequently, we investigated those proteins that localize at the SPB using a constitutively-expressing *atb2* promoter (*Matsuyama et al., 2008*) (*Figure 2A* and subsequent figures, unless otherwise indicated) to simplify experiments. Notably, the localization pattern was the same using both the nmt1 and atb2 promoters.

### *Drosophila* cell culture and transfections

*Drosophila melanogaster* (D.Mel) cells (Thermo Fisher Scientific) were cultured in Express5 SFM (GIBCO) supplemented with 1 × L Glutamine-Penicillin-Streptomycin. DsRNA synthesis was performed as previously described (*Bettencourt-Dias et al., 2005*). Transient plasmid transfections were performed with Effectene reagent (QIAGEN) according to the manual. The primers used to amplify DNA templates for dsRNA synthesis are shown in Key Resources Table. Cells were regularly tested for mycoplasma.

### *Drosophila* stocks and culturing

All the fly stocks used in this study are described in Key Resources Table and publicly available stocks are listed in Flybase (www.flybase.org). Flies were reared according to standard procedures at 25°C on corn meal media (*Jana et al., 2016*).

## Preparation of cell extracts, Western blotting and immunoprecipitation

*S. pombe* lysates were prepared using glass beads in extraction buffer (20 mM Hepes-NaOH (pH 7.5), 50 mM KOAc, 200 mM NaCl, 1 mM EDTA, 0.2% Triton X-100, and 0.1 mM NaF, additionally supplemented with 1 × EDTA free protease inhibitors (Roche) and 1 mM PMSF). Total cell lysates from D.Mel cells were prepared by resuspending cell pellets in lysis buffer described in *Galletta et al. (2014)* (50 mM Tris (pH 7.2), 125 mM NaCl, 2 mM DTT, 0.1% Triton X-100, supplemented with 1 × EDTA free protease inhibitors (Roche) and 1 mM PMSF). Equal amounts of total cell lysates were separated in SDS-PAGE and analyzed by immunoblotting. For the immunoprecipitation, the cell extracts prepared from *S. pombe* or D.Mel cells were incubated for three hours at 4°C with Dynabeads Protein A (Thermo Fisher Scientific) pre-incubated with rabbit anti-GFP (Abcam). The beads were then washed three times with lysis buffer, and boiled in the Laemmli buffer for SDS-PAGE and western blotting.

## Protein purification and in vitro binding assay

*Escherichia coli* strain Rosetta (DE3) was transformed with the expression plasmid, and protein expression was induced at 25°C by the addition of 0.5 mM IPTG overnight. For purification of GST fusion proteins, cell pellets were lysed by sonication in lysis buffer (25 mM Tris pH 7.5, 150 mM NaCl, 1 mM EDTA, and protease inhibitors), and 10% Triton X-100 was added to the lysate at 0.5% final concentration. GST fusion protein was affinity-purified with MagneGST Glutathione Particles (Promega). For purification of 6xHis-PACT, cell pellets were lysed by sonication in lysis buffer (50 mM $NaH_2PO_4$, 300 mM NaCl, 10 mM Imidazole, proteases inhibitors, and 0.1% Tween, pH 8) supplemented with lysozyme at 1 mg/ml final concentration. The protein 6xHis-PACT was affinity-purified with TALON Metal Affinity Resins (Clontech), and eluted with the elution buffer (50 mM $NaH_2PO_4$, 300 mM NaCl, 150 mM Imidazole, proteases inhibitors, 0.1% Tween) and then dialyzed was in the Dialysis Buffer (50 mM $NaH_2PO_4$, 200 mM NaCl, 5 mM Imidazole, pH8).

Recombinant GST or GST-DmSAS-6 beads were incubated with His-PACT supplemented with 500 µl of binding buffer (50 mM Na-HEPES, pH 7.5, 100 mM NaCl, 2 mM MgCl2, 1 mM DTT, 0.1% Triton X-100 and protease inhibitor) at 4°C for 1 hr, and the beads were washed three times in binding buffer. The proteins were eluted in SDS sample buffer and analyzed by SDS-PAGE and western blotting.

## Immunostaining and imaging of *S. pombe* and D.Mel cells

Living or fixed *S. pombe* cells were mounted with a lectin-coated coverslip to immobilize before imaging. Immunostaining of D.Mel cells was performed as previously described (*Cunha-Ferreira et al., 2013*). Cells were mounted with Vectashield containing DAPI (Vector Laboratories). Cell imaging was performed on Nikon Eclipse Ti-E microscopes with Evolve 512 EMCCD camera (Photometrics) or DeltaVision Core system (Applied Precision) inverted microscope (Olympus, IX-71) with Cascade 2 EMCCD camera (Photometrics). Images were acquired as a Z-series (0.3 µm interval) and are presented as maximal intensity projections.

## Structured illumination microscopy (SIM)

All SIM images were collected using Deltavision OMX V3 (GE Healthcare Life Sciences). For SIM images, an oil-immersion Plan-Apo 1.4 NA objective and two different lasers (488 nm and 560 nm) were used and their alignments were corrected using multicolor beads before imaging. All SIM images collected using OMX were reconstructed in OMX SoftWoRx (Deltavision). Subsequently, all images were processed in ImageJ.

## Time-lapse imaging

To capture the localization of SAS-6-GFP in fission yeast, cells were cultured in minimal medium at 30°C, mounted onto a lectin-coated glass-bottomed dish (MatTek). Live cell imaging was performed on Nikon Eclipse Ti-E microscopes with Evolve 512 EMCCD camera (Photometrics) in a chamber maintained at 30°C by a temperature controller (Tokai Hit). Images were acquired as a Z-series (0.3 µm interval) and are presented as maximal intensity projections.

### Immunostaining, imaging and image analysis of *Drosophila* sperm cells

Testes from adult flies were dissected in testes buffer, transferred to poly-L-lysine glass slides, squashed, and snap frozen in liquid nitrogen as previously described (*Jana et al., 2016*). Then, testes were stained using different primary antibodies and secondary antibodies following the published method (*Jana et al., 2016*). Samples were mounted in Vectashield mounting media (Vector Laboratories) and they were examined in microscopes. Given that *Drosophila* has different stages of spermatocytes, we focused on the mature, large G2 spermatocytes and measured the total amount of SAS-6 and PLP at the basal bodies. All Confocal images were collected using Leica TCS SP 5X (Leica Microsystems, Germany) and processed in ImageJ.

### Male fertility tests

Fertility tests were performed by crossing single males with three wild-type females during 3 days. The progeny per tube was scored and averaged for ≥10 males for each genotype.

### Fluorescence intensity quantification and centriole counting

For fluorescence intensity quantification, Z-stack images (0.3 um intervals) of 21 sections (for fission yeast) or 61 sections (for D.Mel cells) were acquired. Integrated intensity of the fission yeast SPB (6 $\times$ 6 pixel) or the centrosome/centriole in D.Mel cells (7 $\times$ 7 pixel) were recorded in the maximum projected images. Cellular background intensity was subtracted from the SPB or centriole intensity. Image processing and quantification were performed in ImageJ.

### Statistical analysis

The statistical analysis (non-parametric Mann Whitney U test) was performed in Graphpad Prism version 5.0 software.

### Bioinformatics analysis to predict orthologs

Complete proteomes were downloaded from Ensembl and Ensembl Genomes databases. EST information was downloaded from JGI. We performed orthology prediction based on multiple methods. Candidate orthologs were identified using pairwise sequence-based (BLASTP and phmmer) and domain-based (hmmsearch) methods (*Altschul et al., 1990*; *Finn et al., 2015*). The orthologs were then manually verified by (1) confirming the presence of critical domains and motifs, (2) using the bidirectional best hit approach. If results were ambiguous (i.e. there are multiple candidates), a putative ortholog was identified by constructing phylogenetic trees of the protein family using MrBayes 3.2.5 (*Ronquist and Huelsenbeck, 2003*). The species tree was downloaded from NCBI taxonomy.

## Acknowledgements

The authors thank E Levy and P Beltrão for critical reading of the manuscript, I Hagan and P Tran for helpful discussions. We are grateful to M Gomes and C Bicho for help with experiments. We thank I Hagan, M Sato, T Toda, K Tanaka, JQ Wu, A Paoletti, K Gould, T Matsumoto, M Bornens, T Megraw, N Dzhindzhev, D Glover, G Rogers, B Galletta, N Rusan, J Raff, RIKEN BRC and NBRP (Japan) for yeast strains, antibodies and plasmids. D Ito is supported by a long-term fellowship from the Human Frontier Science Program (LT000344/2013), and postdoctoral fellowships from the Uehara Memorial Foundation, Japan and Fundação para a Ciência e a Tecnologia (FCT) (195/BPD/17). Work was funded by the Gulbenkian Foundation, an FCT grant (PTDC/BIM-ONC/6858/2014), an FCT investigator grant and a European Research Council Consolidator Grant to MBD (CoG683528).

## Additional information

### Funding

| Funder | Grant reference number | Author |
|---|---|---|
| Human Frontier Science Program | LT000344/2013 | Daisuke Ito |
| Uehara Memorial Foundation | | Daisuke Ito |

| Fundação para a Ciência e a Tecnologia | 195/BPD/17 | Daisuke Ito |
|---|---|---|
| Fundação para a Ciência e a Tecnologia | SFRH/BPD/87479/2012 | Swadhin Chandra Jana |
| Fundação para a Ciência e a Tecnologia | PTDC/BIM-ONC/6858/2014 | Mónica Bettencourt-Dias |
| European Research Council | CoG683528 | Mónica Bettencourt-Dias |
| Calouste Gulbenkian Foundation | | Mónica Bettencourt-Dias |
| Fundação para a Ciência e a Tecnologia | FCT Investigator Grant | Mónica Bettencourt-Dias |

The funders had no role in study design, data collection and interpretation, or the decision to submit the work for publication.

### Author contributions

Daisuke Ito, Conceptualization, Data curation, Formal analysis, Funding acquisition, Validation, Investigation, Visualization, Methodology, Writing—original draft, Writing—review and editing, Created fission yeast strains, Performed fission yeast experiments, Performed bioinformatics analyses, Performed cell biology in D.Mel cells and biochemistry experiments, Performed super-resolution imaging, Conceived the study, Analyzed data; Sihem Zitouni, Data curation, Formal analysis, Investigation, Performed cell biology in D.Mel cells and biochemistry experiments; Swadhin Chandra Jana, Data curation, Formal analysis, Investigation, Performed super-resolution imaging, Performed Drosophila experiments; Paulo Duarte, Data curation, Formal analysis, Investigation, Created fission yeast strains; Jaroslaw Surkont, Data curation, Formal analysis, Investigation, Performed bioinformatics analyses; Zita Carvalho-Santos, Data curation, Formal analysis, Investigation, Created fission yeast strains, Contributed to the conception; José B Pereira-Leal, Miguel Godinho Ferreira, Supervision, Contributed to the conception; Mónica Bettencourt-Dias, Conceptualization, Supervision, Funding acquisition, Writing—original draft, Writing—review and editing, Conceived the study, Analyzed data

### Author ORCIDs

Daisuke Ito https://orcid.org/0000-0002-6759-0901
Swadhin Chandra Jana https://orcid.org/0000-0002-8311-3849
Miguel Godinho Ferreira https://orcid.org/0000-0002-8363-7183
Mónica Bettencourt-Dias https://orcid.org/0000-0003-1987-5598

### Decision letter and Author response

Decision letter https://doi.org/10.7554/eLife.41418.030
Author response https://doi.org/10.7554/eLife.41418.031

## Additional files

### Supplementary files

• Transparent reporting form
DOI: https://doi.org/10.7554/eLife.41418.028

### Data availability

All data generated or analysed during this study are included in the manuscript and supporting files. Source data files are provided for Figure 1, Figures 2, Figure 2-figure supplement 1, Figure 3, Figure 4, Figure 4-figure supplement 1-3, Figure 5, Figure 6, Figure 6-figure supplement 1-2, and Figure 7.

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
