## [Decision Letter]

Thank you for submitting your article "Pericentrin-mediated SAS-6 recruitment promotes centriole assembly" for consideration by *eLife*. Your article has been reviewed by three peer reviewers, and the evaluation has been overseen by a Reviewing Editor and Andrea Musacchio as the Senior Editor. The reviewers have opted to remain anonymous.

The reviewers have discussed the reviews with one another and the Reviewing Editor has drafted this decision to help you prepare a revised submission. As you can see, we are interested in publishing your paper, but we would like you to address some control experiments that we think would greatly strengthen your conclusions.

Ito and colleagues investigated whether *Drosophila* centriole proteins localize to the fission yeast spindle pole body (SPB), and found this to be the case for SAS-6, Bld10 and SAS-4, but not Ana2 and Plk4. Further analysis uncovered that fission yeast Pcp1 is required for SAS-6 SPB localization, and that Pcp1 overexpression generates multiple foci that harbor SAS-6. The authors report that the PACT domain-containing region of Pcp1 mediates interaction with SAS-6, and that the same holds for the PACT domain of *Drosophila* Plp. Furthermore, the authors find in *Drosophila* tissue culture cells that Plp acts in a partially redundant manner with PLK4-STIL in recruiting SAS-6 to centrioles, as well as in promoting centriole formation. The authors then investigate the role of Plp in SAS-6 recruitment and centriole formation during *Drosophila* spermatogenesis. They find that centriole number is not affected by PLPRNAi, but that centriole length is impaired, as reported previously upon SAS-6 depletion. Finally, SAS-6 is shown to interact with the calmodulin binding domain within the PACT domain.

Overall, this is an interesting study that takes advantage in a creative manner of what evolution has to offer, revealing a potentially important facet of the mechanisms regulating centriole formation. However, a few outstanding issues need to be addressed by the authors before publication in *eLife* can be endorsed.

Our main concern is that the authors have not shown that the interaction is direct between Pcp1 and SAS6. They have one experiment using recombinant proteins (showing Figure 5C). That is an IP showing GST-SAS6 pulls down His-PACT. However, there is no quantification of the IP. If GST-SAS6 pulls down nearly all the His-PACT, that is more convincing evidence of a direct interaction. Often in these IPs, though, only 1% of the target protein is pulled down and that is not convincing of a direct interaction. it would be helpful to state what fraction of the total sample is loaded in each well, for the input and the pull down. Then it would be possible to judge the fraction of the protein that is bound. A PACT domain mutant (such has the C911R from Spc110; see Geier et al., 1993) would be useful, particularly in Figure 4D as a control instead of GFP only.

Along these lines, The SIM data reported in Figure 4—figure supplement 1 do not support the claims made in the main text. In the last paragraph of the subsection “SAS-6 interacts with the conserved region of Pcp1, and Pcp1 is sufficient to recruit SAS-6”, the authors conclude that the SIM data supports the notion that SAS-6-GFP associates with Pcp1. However, the data in panel C shows that the distance between SAS-6-GFP and Pcp1-TdT is the same as that between SAS-6-GFP and Sid4-TdT, which is located further away from the nuclear envelope, as schematized also in panel A. Therefore, these data cannot be taken as support that SAS-6-GFP associates with Pcp1 more than it does for instance with Sid4. This is not to say that the data rules out this possibility, but simply that the present analysis by SIM does not allow one to tell one way or the other. The text should be altered accordingly. It would be greatly desirable if additional evidence could be provided that the interaction is indeed direct.

Several of the quantifications need to be conducted in a different manner. It appears that in general the authors report absolute quantifications, for instance of SAS-6-GFP in Figure 2D, and not quantifications relative to another spindle pole body marker, such as Sid4-tdTomato in the case of Figure 2D. Reporting relative quantifications is important to ensure that alterations in signal intensity do not simply reflect differences in the structure of the SPB. A related concern holds for Figure 4—figure supplement 3. Here, the authors take the data as supporting evidence that SAS-6 localization on spindle pole bodies is not due to random binding to a SPB structure. However, levels of Sid4-tdTomato appear to vary as well, calling into question the conclusions drawn from the current quantification. Likewise, the signal intensity per focus reported in Figure 5E should be expressed as a function of the Bld10 signal. Linked to this The authors state that SAS-6 accumulates at the beginning of mitosis, but there seems to be SAS-6 at the unseparated SPBs as well. What do Pcp1 levels look like during these time points? Is this increase a general property of SPB proteins and represents the growth of the second SPB?

There are concerns with the way the *Drosophila* tissue culture cell experiments are analyzed. First, the authors write that cells were arrested in mitosis using colchicine for six hours. One potential concern is that levels of SAS-6 at centrioles may vary during such a long arrest. Analyzing mitotic cells in an unperturbed population would be much preferable. Moreover, no information is given about the concentration of colchicine that was used. Second, after having explained the importance of comparing mitotic cells (subsection “The conserved SAS-6-pericentrin interaction plays a role in centriole assembly”, last paragraph), the authors analyze the number of centrioles in interphase, not in mitosis (in Figure 6D and 6E). Why is this? As stated by the authors themselves, this may potentially lead to confounding effects, for instance if cell cycle progression is altered. Mitotic cells in an unperturbed population must be analyzed instead here as well. The same comment holds for Figure 7, where it appears also that interphase cells were examined. Likewise, mitotic cells should be scored in Figure 5F and 5G. In this case, the legend should also indicate how many cells have been analyzed. Furthermore, instead of simply indicating the fraction of cells with more than 4 centrioles, it would be interesting to show the actual distribution of foci numbers per cell.

This manuscript could benefit from general editing. The text switches between British and American spelling. Some of the results or conclusions are confusing due to wording. For example: "To compare the signal intensity in cells at the same cell cycle stage, mitosis, we introduced both in wild-type and pcp1-14 background the mutation in a mitotic kinesin (cut7-446), which fails in interdigitating the mitotic spindle and causes the cells to arrest in early mitosis.”

[Editors' note: further revisions were requested prior to acceptance, as described below.]

Thank you for submitting your article "Pericentrin-mediated SAS-6 recruitment promotes centriole assembly" for consideration by *eLife*. Your article has been reviewed by three peer reviewers, and the evaluation has been overseen by a Reviewing Editor and Anna Akhmanova as the Senior Editor. The reviewers have opted to remain anonymous.

As you can see, there are a few points that have been raised in review of your revised manuscript. Reading through them, I believe that you can answer them by some textural rewrite.

Essential revisions:

A) The four *Drosophila* proteins that localized to the fission yeast SPB were driven from the constitutive *atb2* promoter, whereas the two that did not were driven from the inducible *nmt1* promoter. Could the use of different promoters have an impact on the localization? Have the authors attempted to swap promoters? At a minimum, further discussion of this experimental difference should be provided. If you have used other promotors, please add this to the text. If not, please explain why you do not see this as a worry.

B) You suggest that time limitations prevented them from "generating a large set of new mutant constructs". Could you be more specific here- please specific why it was hard to make them.

C) You now explain that *Drosophila* possesses centrin1/2-like proteins and quote a book chapter that cannot be accessed on line as supporting evidence (Results, third paragraph). Given the importance of this assertion for the implications of the present study, and also considering the thorough analysis that the authors have put in the bioinformatics analysis presented in Figure 1, these centrin1/2-like proteins should be analyzed using the same methods as the ones used in the present work and included in some fashion in Figure 1. This will enable readers to more fully appreciate whether such proteins are expected to interact with Calmodulin.

I think the best way to deal with this point is to report that this has been done before, and say that you are redoing it with your bioinformatic pipeline for clarity's sake.

D) You indicate that it is difficult to find a centriole component that could be used as an independent marker in the tissue culture experiments reported in Figure 5D and 5E. In this case, the authors should state explicitly in the main text that overexpressing PACT leads to the recruitment not only of SAS-6, but also of Bld10, and perhaps of other centriole proteins. The current writing may give the wrong impression that there is something special about the impact of PACT overexpression on SAS-6 recruitment, whereas it is obvious from Figure 5D that this is also the case for Bld10p at the least.

E) You acknowledge that it is better to analyze cells not treated with colchicine, but assert that it is "technically difficult" to find non-drug treated cells in mitosis. This argument is not serious: a sparsely seeded 6 cm dish will have something like 100'000 cells or more. Even if only 20% are transfected and if only 5% are in mitosis, this would still leave 1000 mitotic transfected cells to score per dish. It seems like this experiment could have been done.

---

## [Author Response]

Overall, this is an interesting study that takes advantage in a creative manner of what evolution has to offer, revealing a potentially important facet of the mechanisms regulating centriole formation. However, a few outstanding issues need to be addressed by the authors before publication in eLife can be endorsed.Our main concern is that the authors have not shown that the interaction is direct between Pcp1 and SAS6. They have one experiment using recombinant proteins (showing Figure 5C). That is an IP showing GST-SAS6 pulls down His-PACT. However, there is no quantification of the IP. If GST-SAS6 pulls down nearly all the His-PACT, that is more convincing evidence of a direct interaction. Often in these IPs, though, only 1% of the target protein is pulled down and that is not convincing of a direct interaction. it would be helpful to state what fraction of the total sample is loaded in each well, for the input and the pull down. Then it would be possible to judge the fraction of the protein that is bound. A PACT domain mutant (such has the C911R from Spc110; see Geier et al., 1993) would be useful, particularly in Figure 4D as a control instead of GFP only.

We thank the reviewer for this question. In our experiment in Figure 5C, 100% of all the samples were loaded in each lane after pull-down. We observed that 56 ± 9% of His-PACT is bound to GST-SAS-6 (three experimental repeats), indicating a direct interaction between PACT and SAS-6. All this information is now mentioned in Figure 5 legend.

We appreciate the suggestion to use a set of PACT domain mutants to better understand the nature of the interaction between SAS-6 and PACT. However, this would demand generating a large set of new mutant constructs within the PACT domain, as they do not exist for *S. pombe*, as they do for *S. cerevisiae*, and their extensive characterization. We were not able to perform these experiments, due to time limitation. This is certainly one of many research avenues opened with the manuscript.

Along these lines, The SIM data reported in Figure 4—figure supplement 1 do not support the claims made in the main text. In the last paragraph of the subsection “SAS-6 interacts with the conserved region of Pcp1, and Pcp1 is sufficient to recruit SAS-6”, the authors conclude that the SIM data supports the notion that SAS-6-GFP associates with Pcp1. However, the data in panel C shows that the distance between SAS-6-GFP and Pcp1-TdT is the same as that between SAS-6-GFP and Sid4-TdT, which is located further away from the nuclear envelope, as schematized also in panel A. Therefore, these data cannot be taken as support that SAS-6-GFP associates with Pcp1 more than it does for instance with Sid4. This is not to say that the data rules out this possibility, but simply that the present analysis by SIM does not allow one to tell one way or the other. The text should be altered accordingly. It would be greatly desirable if additional evidence could be provided that the interaction is indeed direct.

We agree with the reviewer’s concern and have now addressed this limitation by changing the conclusion of the SIM results in the last paragraph of the subsection “SAS-6 interacts with the conserved region of Pcp1, and Pcp1 is sufficient to recruit SAS-6”. Using biochemical (Figure 2, 4) and genetic (Figure 3, 4) approaches, we showed SAS-6-GFP is recruited to the SPB by Pcp1. Though we showed SAS-6 co-localizes with Sid4 and Pcp1 (Figure 2, 4) using epifluorescence micrographs, due to the resolution limit (~200 nm) of a conventional optical microscope, we failed to conclude that SAS-6 is indeed a part of SPB. Thus, using SIM data we found that SAS-6-GFP localizes at ~50 nm distant from both Pcp1-TdT and Sid4-TdT. Given that the diameter and height of fission yeast SPB is 180 nm and 90 nm (Ding et al., 1997), and both Pcp1 and Sid4 are considered as core SPB components, our SIM results suggest that the ectopically expressed SAS-6-GFP localizes to the core of the SPB.

Several of the quantifications need to be conducted in a different manner. It appears that in general the authors report absolute quantifications, for instance of SAS-6-GFP in Figure 2D, and not quantifications relative to another spindle pole body marker, such as Sid4-tdTomato in the case of Figure 2D. Reporting relative quantifications is important to ensure that alterations in signal intensity do not simply reflect differences in the structure of the SPB.

We thank the reviewer for raising this important question. However, it is not clear what would be the ideal spindle pole marker for this quantitation. In fact, we also observed that the signal intensity of Sid4-tdTomato was significantly higher in the *pcp1* mutant at the restrictive temperature (Author response image 1), instead of reduced as predicted by the reviewer. This is perhaps due to some uncharacterized relationship between these molecules. Therefore, Sid4 is not a suitable indicator of the SPB structure in the *pcp1* mutant background and we thus believe that, in these experiments, it would not be appropriate to normalize the SAS-6 amount (or other centriole components) with respect to Sid4, and preferred to show unnormalized data. Importantly, a similar reduction in SAS-6 was observed in *Drosophila*, upon PLP depletion (Figures 6 and 7).

**Author response image 1. respfig1:** Intensity of Sid4 is increased in the pcp1 mutant. (A-F) Quantification of the intensity of the centriole proteins and Sid4-tdTomato per SPB in the indicated conditions. B, D and F correspond to A, C, and E.Means ± s.d. are shown in red (ns-not significant, * p<0.05, ** p<0.001, *** p<0.0001, Mann-Whitney U test).

A related concern holds for Figure 4—figure supplement 3. Here, the authors take the data as supporting evidence that SAS-6 localization on spindle pole bodies is not due to random binding to a SPB structure. However, levels of Sid4-tdTomato appear to vary as well, calling into question the conclusions drawn from the current quantification. Linked to this The authors state that SAS-6 accumulates at the beginning of mitosis, but there seems to be SAS-6 at the unseparated SPBs as well. What do Pcp1 levels look like during these time points? Is this increase a general property of SPB proteins and represents the growth of the second SPB?

We now understand the reviewer’s view and perhaps we were not clear enough in describing the localization behavior of SAS-6-GFP in cycling fission yeast. Previously, we stated that “SAS-6 accumulates at the beginning of mitosis”, but it is incorrect. Now we corrected the sentence to “SAS-6 accumulates at the SPB before entering mitosis”. We apologize for the error.

As correctly pointed by the reviewer, the enrichment of SAS-6-GFP to the SPB before mitosis is positively correlated with that of Sid4 (Figure 4—figure supplement 3D).Furthermore, the behavior of SAS-6-GFP accumulation at the SPB just before mitosis is very similar to that of Pcp1 reported before (Wälde and King, 2014). Therefore, SAS-6-GFP enrichment at the SPB seems to correlate with the growth and maturation of the SPB, with Pcp1 accumulation. We have included this modified description in the main text. Please check the last paragraph of the subsection “Regulation of SAS-6 localization on SPB”.

Likewise, the signal intensity per focus reported in Figure 5E should be expressed as a function of the Bld10 signal.

We understand the reviewer’s concern, but a recent study from our group (Jana et al., 2018) demonstrated that SAS-6 and Ana2 are required for Bld10 localization to the centriole. Therefore, Bld10 is not an independent marker and not a suitable indicator of the centriole size. We think that it is not appropriate to normalize SAS-6 intensity with respect to Bld10 in Figures 5 and 6. Moreover, given that SAS-6 is a critical component of the procentriole, deregulation of SAS-6 at a given centriole would change its composition. Therefore, it is difficult to find a centriole component, which could be used as an independent marker in these experiments.

There are concerns with the way the *Drosophila* tissue culture cell experiments are analyzed. First, the authors write that cells were arrested in mitosis using colchicine for six hours. One potential concern is that levels of SAS-6 at centrioles may vary during such a long arrest. Analyzing mitotic cells in an unperturbed population would be much preferable. Moreover, no information is given about the concentration of colchicine that was used.

We agree with the reviewer’s concern. Although it is better to use untreated cells, in D.Mel cells, the transfection efficiency is usually about 20-30% and the mitotic index is around 5%, therefore finding transfected cells in mitosis is technically difficult. Mitotic cells were thus enriched by colchicine-induced M arrest, with controls being treated the same way (so far there is no indication in the literature that PLP depletion would impair or enhance mitotic progression in *Drosophila* cultured cells-see for example (Goshima et al., 2007). We treated cells with 12.5μM of colchicine for 6 hours. We have now added this information in the legend of Figure 5.

Second, after having explained the importance of comparing mitotic cells (subsection “The conserved SAS-6-pericentrin interaction plays a role in centriole assembly”, last paragraph), the authors analyze the number of centrioles in interphase, not in mitosis (in Figure 6D and 6E). Why is this? As stated by the authors themselves, this may potentially lead to confounding effects, for instance if cell cycle progression is altered. Mitotic cells in an unperturbed population must be analyzed instead here as well. The same comment holds for Figure 7, where it appears also that interphase cells were examined. Likewise, mitotic cells should be scored in Figure 5F and 5G. In this case, the legend should also indicate how many cells have been analyzed.

We thank the reviewer for pointing out this inconsistency. In addition to the quantification of SAS6 intensity at the centrioles, we have now analysed the number of centrioles in mitotic cells upon depletion of Ana2, PLP and both. Importantly, the defects in centriole numbers are well correlated to the SAS-6 localization defects observed in Ana2 and PLP depleted situations. Moreover, we found that, in RNAi treated conditions, the centriole number defects observed in interphase cells (Figure 6—figure supplement 1) are very similar to the defects observed in mitotic cells (Figure 6D, E). The latter result suggests that the effect of deregulation of centrosome components can be monitored in both interphase and mitosis. Therefore, in PACT overexpression (Figure 5) and PLP rescue (Figure 6—figure supplement 2) experiments, we analysed the centriole number in interphase as it is difficult to image a large number of mitotic cells in transfection experiments as discussed above.

Following the reviewer’s suggestion, we have now added the number of cells analysed for counting centriole numbers in the respective figure legends.

Furthermore, instead of simply indicating the fraction of cells with more than 4 centrioles, it would be interesting to show the actual distribution of foci numbers per cell.

We thank the reviewer for the suggestion. We have now replaced the graph with a new one showing the distribution of centriole numbers in Figure 5G.

This manuscript could benefit from general editing. The text switches between British and American spelling. Some of the results or conclusions are confusing due to wording. For example: "To compare the signal intensity in cells at the same cell cycle stage, mitosis, we introduced both in wild-type and pcp1-14 background the mutation in a mitotic kinesin (cut7-446), which fails in interdigitating the mitotic spindle and causes the cells to arrest in early mitosis.”

We thank the reviewer for the suggestion. Please check the modified text in the first paragraph of the subsection “Fission yeast Pcp1 is required for SAS-6 recruitment”. Other grammatical errors have also been corrected in the text.

[Editors' note: further revisions were requested prior to acceptance, as described below.]

Thank you for submitting your article "Pericentrin-mediated SAS-6 recruitment promotes centriole assembly" for consideration by eLife. Your article has been reviewed by three peer reviewers, and the evaluation has been overseen by a Reviewing Editor and Anna Akhmanova as the Senior Editor. The reviewers have opted to remain anonymous.As you can see, there are a few points that have been raised in review of your revised manuscript. Reading through them, I believe that you can answer them by some textural rewrite.Essential revisions:A) The four *Drosophila* proteins that localized to the fission yeast SPB were driven from the constitutive atb2 promoter, whereas the two that did not were driven from the inducible nmt1 promoter. Could the use of different promoters have an impact on the localization? Have the authors attempted to swap promoters? At a minimum, further discussion of this experimental difference should be provided. If you have used other promotors, please add this to the text. If not, please explain why you do not see this as a worry.

We thank the reviewer for pointing this out. We have indeed used *nmt1* promoters for all the *Drosophila* centriole genes studied in this manuscript. Notably, the localization pattern was the same using both the *nmt1* and *atb2* promoters. We have now mentioned this information in the Materials and methods section (subsection “Expression of *Drosophila* centriole proteins in fission yeast”). In the manuscript, we focused on how the localization of the three proteins on the SPB is regulated using the constitutively-expressing *atb2* promoter in order to simplify the experiments (no induction needed; Figure 2A).

B) You suggest that time limitations prevented them from "generating a large set of new mutant constructs". Could you be more specific here- please specific why it was hard to make them.

We thank the reviewer for raising this point, and we do apologize for not having explained well the problems associated with this experiment in our previous rebuttal. The reviewer wrote, “a PACT domain mutant (such as the C911R from Spc110; see Geier et al., 1993) would be useful, particularly in Figure 4D as a control instead of GFP only.“

Though the budding yeast (*S. cerevisiae*) Spc110-C911R mutant is known to be deficient in binding to calmodulin (Sundberg et al., 1996), we believe that at this stage, the suggested mutation would not be the ideal control for our experiment for the following reasons:

1) Until we characterize the critical fission yeast Pcp1 residues in the PACT domain that are required to bind to *Drosophila* SAS-6, the suggested mutant might not be the best negative control.

2) The residue C911 in Spc110 (*S. cerevisiae*) is not conserved in fission yeast (Flory et al., 2002).

Future work aimed at further understanding the evolution of the SAS-6-PACT interaction, should ideally get the 3D structure of the interacting surfaces and then test for PACT mutants that are deficient in interacting with SAS-6 in vitro and in vivo.

C) You now explain that *Drosophila* possesses centrin1/2-like proteins and quote a book chapter that cannot be accessed on line as supporting evidence (Results, third paragraph). Given the importance of this assertion for the implications of the present study, and also considering the thorough analysis that the authors have put in the bioinformatics analysis presented in Figure 1, these centrin1/2-like proteins should be analyzed using the same methods as the ones used in the present work and included in some fashion in Figure 1. This will enable readers to more fully appreciate whether such proteins are expected to interact with Calmodulin.I think the best way to deal with this point is to report that this has been done before, and say that you are redoing it with your bioinformatic pipeline for clarity's sake.

We thank the reviewer for raising this critical point. We have detected a putative Cdc31/centrin3 ortholog in *Drosophila* (FBpp0080611) by the bi-directional BLAST approach. We, however, did not initially value this result, in face of the closer relationship of FBpp0080611 with centrin1/2 (Azimzadeh and Bornens, 2005). Re-examination of this, as per the reviewer’s suggestions, led us to the conclusion that the distance diagram presented in (Azimzadeh and Bornens, 2005) is not sufficient to invalidate the claim that the *Drosophila* gene shares a common ancestor with cdc31 and that they are likely orthologues as the bi-directional BLAST approach indicates. We apologize for this error in the interpretation of the data in the previous version of the manuscript and have now modified Figure 1B and the text accordingly (Results, third paragraph).

Furthermore, perhaps we were also not clear in our previous rebuttal in describing the differential roles of centrins and pericentrin known from the literature. Importantly, centrin is not described as a calmodulin-binding protein, but a calcium-binding protein (Salisbury, 1995), while pericentrin, the central molecule in this manuscript, is a calmodulin-binding protein.

D) You indicate that it is difficult to find a centriole component that could be used as an independent marker in the tissue culture experiments reported in Figure 5D and 5E. In this case, the authors should state explicitly in the main text that overexpressing PACT leads to the recruitment not only of SAS-6, but also of Bld10, and perhaps of other centriole proteins. The current writing may give the wrong impression that there is something special about the impact of PACT overexpression on SAS-6 recruitment, whereas it is obvious from Figure 5D that this is also the case for Bld10p at the least.

We thank the reviewer for raising this point. We have now described in the text the possibility that PACT overexpression also impacts the recruitment of other components, directly or indirectly (subsection “The conserved SAS-6-pericentrin interaction plays a role in centriole assembly”, last paragraph).

E) You acknowledge that it is better to analyze cells not treated with colchicine, but assert that it is "technically difficult" to find non-drug treated cells in mitosis. This argument is not serious: a sparsely seeded 6 cm dish will have something like 100'000 cells or more. Even if only 20% are transfected and if only 5% are in mitosis, this would still leave 1000 mitotic transfected cells to score per dish. It seems like this experiment could have been done.

We apologize for not being clear on why we chose this experimental strategy. The aim of this experiment was to test whether overexpression of PACT leads to more recruitment of *endogenous* SAS-6 and increase in centriole numbers. Given the low amounts of SAS-6 observed in interphase in D.Mel cells, and that SAS-6 levels increase as interphase progresses and are higher in mitosis (our own unpublished observations; see also recent paper (Aydogan et al., 2018)), we scored its levels in mitosis. This ensured a homogeneous population regarding SAS-6 levels, with high levels, allowing for better quantitation. Previously, it had been shown that colchicine treatment for 8 hours does not interfere with centrosome maturation and centriole duplication (Dobbelaere et al., 2008), thus facilitating the finding and therefore analysis of mitotic cells. Goshima and collaborators (Goshima et al., 2007) also arrested cells in mitosis (by targeting APC/C) to analyse multiple phenotypes, including those related to centrosome duplication and maturation. In our case, we used lower concentration and shorter incubation with colchicine (half the amount and 6 hours incubation as compared to (Dobbelaere et al., 2008)), to enrich for mitotic, transfected cells. We now refer to those papers in the main text (subsection “The conserved SAS-6-pericentrin interaction plays a role in centriole assembly”, last paragraph), where this strategy was used and no perturbation was observed in reading out centriole duplication and maturation.